# On the Effect of Pre-training for Transformer in Different Modality on Offline Reinforcement Learning

**Shiro Takagi** *
Independent Researcher
takagi4646@gmail.com

## Abstract

We empirically investigate how pre-training on data of different modalities, such as language and vision, affects fine-tuning of Transformer-based models to Mujoco offline reinforcement learning tasks. Analysis of the internal representation reveals that the pre-trained Transformers acquire largely different representations before and after pre-training, but acquire less information of data in fine-tuning than the randomly initialized one. A closer look at the parameter changes of the pre-trained Transformers reveals that their parameters do not change that much and that the bad performance of the model pre-trained with image data could partially come from large gradients and gradient clipping. To study what information the Transformer pre-trained with language data utilizes, we fine-tune this model with no context provided, finding that the model learns efficiently even without context information. Subsequent follow-up analysis supports the hypothesis that pre-training with language data is likely to make the Transformer get context-like information and utilize it to solve the downstream task.

## 1 Introduction

The past few years have witnessed the tremendous success of pre-trained Transformer-based models [1] on versatile natural language processing (NLP) tasks. This is because pre-training with a large corpus enables Transformer-based models to gain rich universal language representations transferable to vast downstream tasks [2, 3, 4]. Remarkably, recent studies demonstrate that Transformers pre-trained with language data can efficiently solve even non-NLP tasks [5, 6, 7, 8, 9, 10]. Because of its versatility, it has been suggested that Transformers pre-trained with a large corpus could be called a kind of *universal computation engines* [5].

One area worthwhile attempting to study the benefits of pre-trained Transformer is reinforcement learning because the applicability of pre-training to this area still has room to be explored [11, 12, 13]. In this vein of research, Reid et al. studied the effect of pre-training of Transformers on offline reinforcement learning (offline RL) tasks [9]. Their study demonstrates a surprising contrast that pre-training with image significantly deteriorates performance, while that with language data does not negatively affect downstream tasks, rather it even improves performance for some datasets.

However, it is not yet known how the presence or absence of pre-training and its content leads to differences in the performance of downstream tasks. What information does/doesn't the model pre-trained on language data (*language-pre-trained model*) leverage to solve downstream tasks, and why does the model pre-trained on image data (*image-pre-trained model*) result in catastrophic performance? In this paper, we approach these questions by analyzing the inside of the Transformer

---

*https://t46.github.io/

Code is available at https://github.com/t46/pre-training-different-modality-offline-rl.

LaTeX source is available at https://github.com/t46/paper-pre-training-different-modality-offline-rl.

that is fine-tuning to Mujoco [14] offline RL tasks. Analyzing the internals of Transformer, we study how randomly initialized, language-pre-trained, and image-pre-trained models acquire different representations, and what the differences could be attributed to. Our contributions are as follows:

- We study the internal representation of pre-trained and randomly initialized Transformers and find that the pre-trained models largely change their representation, while they encode less information of data during fine-tuning.

- The analysis of parameter change and gradient shows that the pre-trained models do not change parameters much perhaps due to gradient confusion and that the performance of the image-pre-trained model might be partly affected by gradient clipping on the large gradient.

- We examine context-dependence by fine-tuning models with giving no context information, finding that the language-pre-trained Transformer efficiently learns even without context and this could be the case by replacing just a single Transformer block with a pre-trained one.

- Subsequent analysis of the change in how far away the model process the tokens of input supports the possibility that some contextual equivalent information has been positively transferred from language pre-training.

## 2  Related Works

The study of transferability of neural representation has been one of the most important research topics in machine learning research. A bunch of studies have investigated how pre-training influences the downstream tasks, such as computer vision tasks [15, 16, 17, 18] and NLP tasks [19, 20, 21]. Recently, several studies have investigated the transferability of Transformers to various NLP tasks since understanding its high transferability is of great interest. Previous research reveals that Transformer representation encodes some syntax [22], semantics [23], cross-lingual information [24], and world knowledge [25]. Also, it has been pointed out that Transformers obtains not only language-specific information but also more abstract inter-token dependencies [26, 27, 28]. This finding suggests that pre-trained Transformers may apply to a wider variety of tasks than just NLP tasks.

In recent years, studies have emerged that apply the Transformer to tasks other than NLP tasks, reporting the versatility of the Transformer even in such multi-modal cases. Such prior works have shown that language-pre-trained Transformers can efficiently learn image classification [5], symbolic mathematics [6], simple arithmetic [4], reinforcement learning tasks [7, 8, 10, 9], and more generally, sequence modeling problems [5]. Our study is in line with the research that examines the applicability of pre-trained Transformers on reinforcement learning.

Most prior works applying pre-trained Transformer on reinforcement learning explicitly express goals, observations, or actions in a language [29, 8, 10]. On the other hand, the work of [9] applies pre-trained Transformers on the offline RL that language is unrelated to (e.g. Mujoco [14]) and still reports that pre-training is helpful. This study is interesting because it suggests that language-pre-trained Transformers seem to exploit some similarity between the structure of natural language data and trajectory data, which is more abstract and fundamental than linguistic information. Also, exploring the applicability of pre-trained Transformers, which have already been successful, is of great engineering significance since pre-training in offline RL has room for improvement compared to other modalities. Examining the internals of Transformers, we further delve into the findings from this study by Reid et al. [9] and explores the applicability of Transformers to offline RL.

## 3  Background

### 3.1  Transformer

Transformers are sequence-to-sequence models composed of stacked identical blocks, called *Transformer blocks*. Each block consists of self-attention [1], multi-layer perceptron, skip-connection [30], and layer normalization (layer norm) [31]. Denoting the linear projection of input sequence of length $n$ to query, key, values of length $m$ by $Q \in \mathbb{R}^{n \times d_Q}$, $K \in \mathbb{R}^{m \times d_K}$, and $V \in \mathbb{R}^{m \times d_V}$, respectively, the self-attention is $\text{Attention}(Q, K, V) = \text{softmax}\left(\frac{QK^\top}{\sqrt{d_K}}\right) V$. We use GPTs [32], and hence the autoregressive language model, where queries head to only keys before their position.

### 3.2 Offline Reinforcement Learning and Decision Transformer

Reinforcement learning is the problem to learn to make optimal decisions in a dynamic environment. For ease of mathematical handling, the environment is usually described as a Markov decision process (MDP). The MDP consists of states $s \in \mathcal{S}$, actions $a \in \mathcal{A}$, transition probability $P(s'|s, a)$, and reward function $r(s, a) : \mathcal{S} \times \mathcal{A} \to \mathbb{R}$. The interaction of the agent in the environment is represented as $K$ length trajectory, or sequence of state, action, and reward $(s_0, a_0, r_0, s_1, a_1, r_1, ..., s_K, a_K, r_K)$, where $s_t$, $a_t$, and $r_t$ are state, action, and reward at time step $t$. The goal is to find an optimal policy $\pi(a|s)$ to maximize the expected cumulative rewards $\mathbb{E}[\sum_{t=0}^{K} r_t]$ for the trajectory.

Offline RL aims to achieve objectives of reinforcement learning from the trajectory data collected by some policy [33]. Because offline RL is purely characterized by trajectory data, several studies have proposed formulating offline RL as a sequence modeling problem [34, 35]. Decision Transformer is a seminal work that attempts to use a causal transformer (in other words, GPT architecture or decoder of BERT [2]) to solve such a sequence modeling problem [35]. In Decision Transformer, input data is a sequence $\tau = (\hat{R}_0, s_0, a_0, \hat{R}_1, s_1, a_1, ..., \hat{R}_K, s_K, a_K)$ of length $K$, where return-to-go $\hat{R}_t = \sum_{i=t}^{K} r_i$ is the cumulative reward from time step $t$. The previous studies [9, 35] employ the above problem set up to discuss the applicability of Transformer for offline RL. We also follow these studies and use the setup for our analysis.

## 4 Experimental Setup

Unless otherwise noted, all of our experimental setups, including model configuration, dataset, and hyperparameters follow the previous work [9] [2]. The sanity check that our trained models achieve comparable performance with the previous work is shown in Table 1. Details are in Appendix A.

We compare the model with GPT2 architecture [3] that is randomly initialized (*randomly initialized model*), pre-trained with language data (*GPT2*), and with image data (*iGPT*) to study the effect of pre-training. Pre-trained models are from the `Transformers` library from HuggingFace [36]. Like the previous study, the model code for GPT2 and iGPT are `gpt2` and `openai/imagegpt-small`.

For offline RL tasks, we use some of Mujoco tasks [14] in OpenAI Gym [37] provided by D4RL [38] [3], an offline RL dataset library. Specifically, we employ *medium* datasets of *HalfCheetah*, *Walker2d*, and *Hopper* environments. Unless otherwise specified, we show the result for *Hopper-medium*. We put the results for other environments in Appendix J.

Following the previous study [9], we train the models for 40 epochs, each of which consists of 2500 steps. In the following sections, we refer to the 40th epoch checkpoint as the *post-fine-tuning model* and the 0th epoch model (the initial state) as the *pre-fine-tuning model*. The training details, including hyperparameters and the optimizer, are detailed in Appendix B.

**Table 1:** Normalized mean return.

| Dataset | Environment | GPT2 ([9]) | iGPT ([9]) | DT ([9]) | GPT2 (ours) | iGPT (ours) | Random Init (ours) |
|---------|-------------|------------|------------|----------|-------------|-------------|--------------------|
| Medium | Hopper | $79.1 \pm 1.1$ | $5.7 \pm 1.5$ | 67.6 | $79.5 \pm 1.3$ | $2.6 \pm 0.3$ | $67.7 \pm 3.0$ |
| | HalfCheetah | $42.8 \pm 0.1$ | $1.5 \pm 0.1$ | 42.6 | $48.4 \pm 0.0$ | $1.3 \pm 0.1$ | $48.6 \pm 0.2$ |
| | Walker 2D | $78.3 \pm 1.5$ | $0.4 \pm 0.4$ | 74.0 | $71.2 \pm 1.3$ | $4.3 \pm 1.9$ | $71.0 \pm 0.5$ |

## 5 Results and Analysis

### 5.1 Activation Similarity

We first study how the pre-trained and randomly initialized models shape the representation in fine-tuning. To that end, we compare the centered kernel alignment (CKA) [39] of each layer's activation between pre and post-fine-tuning, investigating how each layer's representation changes by learning

---

[2]We use the following code for experiments: https://github.com/machelreid/can-wikipedia-help-offline-rl
[3]https://github.com/rail-berkeley/d4rl

offline RL data [4]. The CKA has been used in various studies and has provided numerous insights into the understanding of neural networks [40, 41, 42, 43, 44]. In particular, we compute the linear CKA, following previous studies [39, 45], for a subset of the dataset. We use the activation of each layer from the last time step of the context for return-to-go, state, and action, respectively. We show the result of state, putting the results for the others in Appendix J.1. Following the previous study [43], we compute CKA of activation not only for Transformer block output but for all internal activation of the Transformer blocks. The definition of the CKA and the details are in Appendices D.2 and D.1.

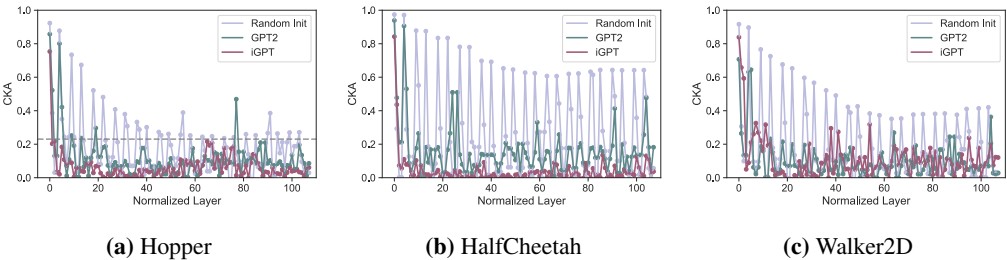

**(a)** Hopper        **(b)** HalfCheetah        **(c)** Walker2D

**Figure 1:** CKA similarity of each layer between pre and post-fine-tuning.

Fig. 1 shows the result. The y-axis is each layer's CKA between representation at pre and post-fine-tuning. The x-axis is the normalized layer index: because the number of layers of iGPT (216) is twice the others (108), we average CKA over two adjacent elements starting from the 0th layer for iGPT and align the x-axis for different models for comparison. Higher CKA means higher similarity.

We observe that the CKAs of pre-trained models (GPT2 and iGPT) are uniformly low across layers except for a few layers, indicating that pre-trained models largely change representation. On the other hand, for the randomly initialized model (Random Init), the representations have a higher similarity to the initial state from the middle to the shallow layers. Also, the CKAs of the randomly initialized model seem to be higher than pre-trained models across layers. These observations indicate that the randomly initialized and pre-trained models structure their representations in very different ways. Further support for this conclusion is shown in Appendix D.4.

To identify which layers have relatively higher CKA for the randomly initialized model, we obtain layer names with CKA values above a threshold (dashed line at 0.23 in Fig. 1 (a)). Then, we find that most of them are layer normalization layers (the list of the layer names are put in Appendix D.3). Therefore, we can speculate that the difference in CKA similarity of layer normalization can be related to the effect of pre-training. This possibility could be supported by previous research [5] that reports that layer norm parameters are the most important to fine-tune.

In uni-modal cases, where the statistical nature of the input data does not change significantly, previous studies have a consensus that the shallow layer acquires a more universal representation and the deep layer acquires a task-specific representation for a wide range of architectures and tasks [46, 47, 48, 49, 50, 39, 51, 42, 44]. The finding that the language-pre-trained Transformer performs as well as or better than the baseline but largely changes the representation suggests that in multi-modal cases, something different may be happening in forming the representation than in prior studies.

## 5.2 Mutual Information Between Hidden Representation and Data

In the previous section, we observe that pre-trained models drastically change their internal representation during fine-tuning, attaining largely different representations. One possible consequence of this observation is that the pre-trained model may have adapted better to the data in the downstream tasks. To explore this possibility, we calculate estimated values of mutual information between hidden representation and data and compare them among different models. Mutual information is a well-known measure of the mutual dependence between the two random variables. Many previous studies have used this metric to investigate the extent to which neural representation encodes input and label information and provided profound insight into how neural networks work [52, 53, 54, 55, 56].

---

[4]We use the following code to compute CKA: https://github.com/google-research/google-research/tree/master/representation_similarity [39]

In particular, we calculate the estimated mutual information $\hat{I}(X;T)$ between some layer representation $T$ and input data $X$, and that $\hat{I}(Y;T)$ between $T$ and label $Y$. We employ Mutual Information Neural Estimation (MINE) [57] to estimate mutual information since this method can estimate mutual information even when random vectors are continuous and have different dimensions [5]. Since Decision Transformers are trained to minimize the gap between predicted action $\hat{a}_t$ and actual action $a_t$, we define the label as $Y = a_t$. Unlike non-sequential models, the causal Transformer exploits past context information. Thus, denoting the input token except for $a_K$ by $x_t$ and the internal activation of Transformer block $l$ of the input by $T_l(x_t)$, we compute $\hat{I}(x_t; T_l(s_K))$ as $\hat{I}(X;T)$ and $\hat{I}(a_K; T_l(x_t))$ as $\hat{I}(Y;T)$ for all token positions, $t = 0, ..., K$. We calculate these values for the shallow, middle, and deep Transformer blocks, respectively. We show the result of the middle block here and leave the remaining in Appendix J.2. The other details are described in Appendix E.1.

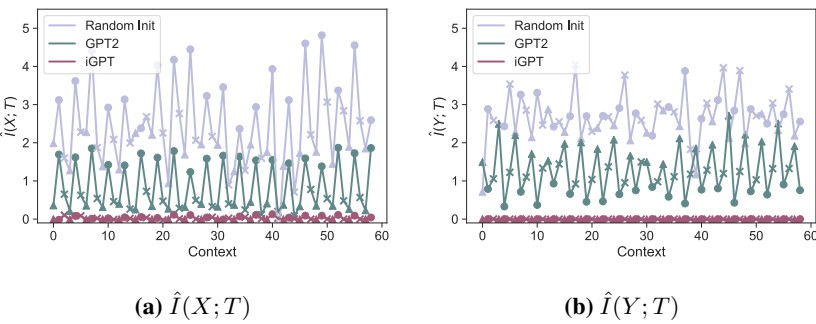

**(a)** $\hat{I}(X;T)$          **(b)** $\hat{I}(Y;T)$

**Figure 2:** Estimated mutual information between data and hidden representation.

Fig. 2 is the result, where Fig. 2 (a) and (b) are results of $\hat{I}(X;T)$ and $\hat{I}(Y;T)$, respectively. The markers *triangle*, *circle*, and *cross* represent return-to-go, state, and action, respectively. The x-axis is the position of tokens in the context and the y-axis is the estimated value of mutual information.

We observe that the randomly initialized model consistently has higher mutual information both for input and label than pre-trained models. Given the catastrophic performance of iGPT, it is understandable that iGPT representations have almost no information. The result comparing GPT2 to random initialization indicates that the changed representation observed in Section 5.1 does not encode more information about the input or the label than the randomly initialized model. That is, the language-pre-trained model does well probably not because it acquires as much or more information about the data as the random initialization. This result suggests that although the representation has changed, there remains something unchanged, and the language-pre-trained model may leverage it to solve the offline RL task. Note that this might be affected by limitations of mutual information.

We also notice that for label the language-pre-trained model has consistently higher mutual information for a specific token type (return-to-go), while that is not always the case for the randomly initialized model. This result implies that the language-pre-trained model may process the information of a specific token type at each layer for prediction.

## 5.3 Parameter Similarity

In the previous section, we discuss the possibility that the language-pre-trained Transformer exploits some pre-acquired information. To investigate this issue on a finer scale, we turn our eye from representation to parameter. We conduct parameter-level similarity analysis and investigate to what extent parameter change between pre and post-fine-tuning. In concrete, we compute $l2$ distance and cosine similarity of parameters between pre and post-fine-tuning. The details are in Appendix F.1.

The result for $l2$ distance is shown in Fig. 3 and for cosine similarity in Fig. 4, where the top is the result for GPT2, the middle is for iGPT, and the bottom is for random initialization. The x-axis is the parameter set index, where, from left to right, the ticks correspond to the shallowest to deepest layer parameters. For visibility, we do not show the labels for ticks and put them in Appendix F.2.

---

[5]We use the following code for mutual information estimation: https://github.com/gtegner/mine-pytorch

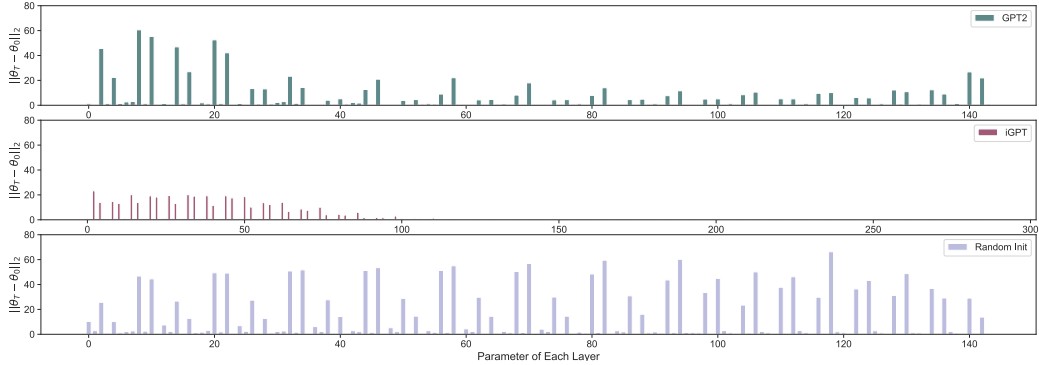

**Figure 3:** $l2$ distance of each parameter between pre and post-fine-tuning.

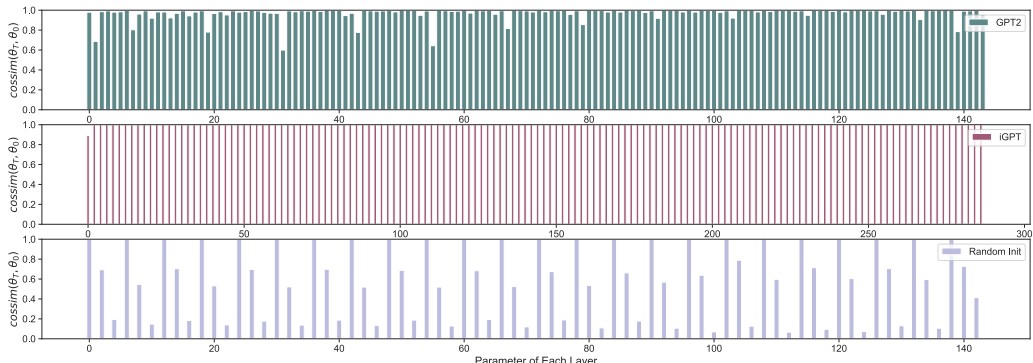

**Figure 4:** Cosine similarity of each parameter between pre and post-fine-tuning.

Overall, We observe that $l2$ distance is small and cosine similarity is large for pre-trained models, meaning that the parameters of pre-trained models do not change that much. At first glance, this appears to contradict the results of Section 5.1. However, prior research has reported that most of the information in Transformers propagates through skip connections [43]. This means that, as long as the parameters of the shallow layers change, the representation of each layer may change significantly without changing the parameters of each layer. We indeed observe that the shallow layer $l2$ distance of the pre-trained model shows relatively large changes. The existence of unchanged parameters means that knowledge acquired through pre-training could be stored in those parameters.

## 5.4 Gradient Analysis

In Section 5.3, we observe that pre-trained models' parameters don't change that much. We dig into this issue, investigating why GPT2 and iGPT do not change parameters. To this end, we compare the gradient norm and *gradient confusion* [58] at the early phase of the training (1st epoch), discussing the training difficulty. When parameter gradients of loss for two different samples are negatively correlated, we say that there is gradient confusion. Empirically, we can measure this by the minimum gradient cosine similarity among all pairs of gradients: when the value is close to 0, gradient confusion is low. The previous study showed that training is easier when gradient confusion is low [58]. Because gradient clipping is used in the previous work [9] and this study (the maximum norm is 0.25), we compute the values for clipped gradients. The details are described in Appendix G.1.

Fig. 5 is the result for the gradient confusion and Fig. 6 is that for gradient norm. Fig. 5 shows that the minimum cosine similarity of iGPT and GPT2 is similarly smaller than the randomly initialized model. This high gradient confusion probably makes pre-trained models harder to train, which explains why they change parameters less than the randomly initialized model. Fig. 6, on the other hand, shows that the gradient norm of iGPT is concentrated around the clipping threshold (0.25), while that of the others is below the bound. That is, the differences in the magnitude of the gradient are collapsed for iGPT, reducing the informational value of the gradients.

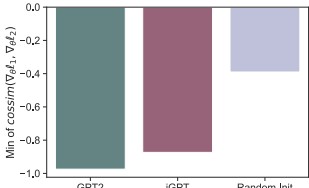

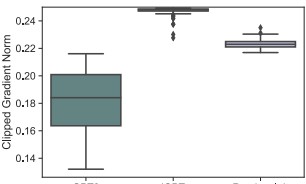

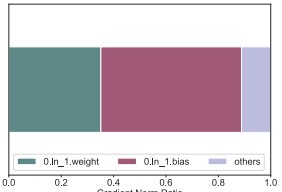

**Figure 5:** Grad. confusion.    **Figure 6:** Grad. norm.    **Figure 7:** iGPT's grad. norm ratio.

To identify which parameter of iGPT dominates the gradient norm, we compare the gradient norm of each parameter set. We calculate the gradient norm ratio of weights and biases of all parameters in all Transformer blocks. The result is shown in Fig. 7; for visibility, we show the parameters of the first layer norm module (weights `0.ln_1.weight` and biases `0.ln_1.bias`) and lump the remaining results together as `others`. The full result is shown in Appendix G.2. We observe that weights and biases of the first layer norm module dominate the norm. This result implies that the difficulty of training iGPT might partially come from the use of gradient clipping on large gradients dominated by few parameters. A complementary analysis is described in Appendix G.3.

## 5.5 Fine-Tuning with No Context Information

In the sections up to this point, we find the possibility that the language-pre-trained model may exploit some pre-acquired information. Then what information does the language-pre-trained model exploit to solve the tasks? One possibility is the ability to handle context because Transformer is skilled at handling the relations of tokens, which enables it to perform well in a variety of NLP tasks [22]. Therefore, we train the language-pre-trained and the randomly initialized model with no access to the context ($K = 1$) to dig into this possibility. If a model learns efficiently without context, the model is already likely to have context-equivalent information. Details of experiments are in Appendix H.1.

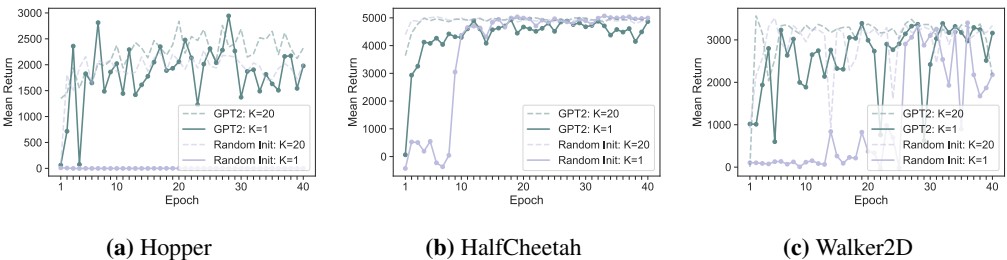

**(a)** Hopper    **(b)** HalfCheetah    **(c)** Walker2D

**Figure 8:** Mean return throughout fine-tuning when access to the context information is prohibited.

The mean return throughout training time is shown in Fig. 8 and the normalized scores of return [6] are shown in Table 2. The solid lines in Fig. 8 are results without context and the dashed lines for reference are those with context. From Fig. 8, we observe that when the context is not provided ($K = 1$), the GPT2 reaches the high mean return much faster than the randomly initialized model. Furthermore, Table 2

**Table 2:** Normalized return of $K = 1$.

| Dataset | Environment | GPT2 | Random Init |
|---------|-------------|------|-------------|
| Medium | Hopper | 83.73 | $-0.24$ |
| | HalfCheetah | 47.7 | 49.0 |
| | Walker 2D | 69.0 | 69.1 |

reveals that the randomly initialized model fails to learn the trajectory for a dataset (*Hopper*), while the language-pre-trained model consistently achieves a comparable performance with that of the baseline result ($K = 20$). That is, we observe that the language-pre-trained Transformer consistently performs more efficiently when no context is provided. This is somewhat surprising, given that previous studies have pointed out that Decision Transformer performs significantly worse with no context [35]. The result that the language-pre-trained Transformer learns efficiently as well without

---

[6]Following the previous studies [38, 35, 9], we report normalized scores: $100 \times \frac{\text{score} - \text{random score}}{\text{expert score} - \text{random score}}$.

context as it does with context suggests that the model may have already acquired some kind of context-like information through pre-training and is utilizing it to solve the task.

## 5.6 More In-Depth Analysis of Context Dependence

### 5.6.1 Replacement by the Pre-Trained Block

In the previous sections, we find that the language-pre-trained Transformer exploits context-related information to solve the task. To identify which block contains the useful information, we conduct a block replacement experiment, where we replace a Transformer block of the randomly initialized model with that of the language-pre-trained model and then fine-tune the model without context. If replacement of a block does not help training, the block probably did not learn effective information in pre-training. Because we observe in Section 5.5 that the early phase characterizes the difference of the models, we focus on the epochs up to 10th epoch. We explain the other details in Appendix I.1.1.

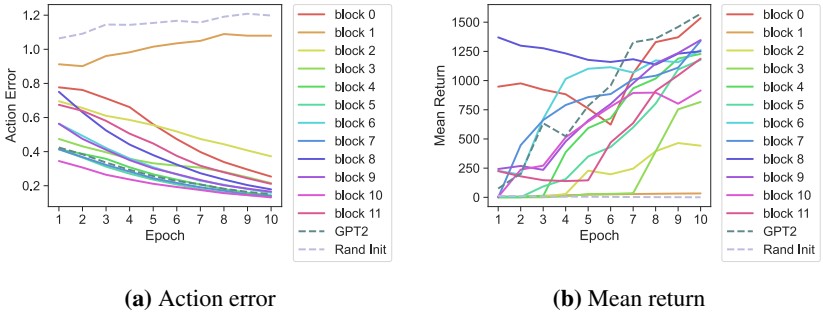

**(a)** Action error                          **(b)** Mean return

**Figure 9:** Learning curve when a single Transformer block is replaced with a pre-trained one.

The result is shown in Fig. 9, where (a) is the prediction error of action and (b) is the mean return [7]. To get a rough trend, we take the exponential moving average. We observe that only when block 1 is replaced with the pre-trained block, does the action error (Fig. 9 (a)) and the mean return (Fig. 9 (b)) behave similarly to the randomly initialized model. Thus, we conclude that all blocks other than block 1 contain context-equivalent useful information to solve tasks. The findings that replacing a single block with the pre-trained one improves learning efficiency and that effective information is distributed across layers alone are surprising enough. This could be related to the dominant role of skip connections for information propagation [43] or the nature of non-monotonic change in the content each layer processes [20] in Transformers. We show the result for Hopper since the trend explained above is most pronounced. For other environments, although the similarity between the random model and block1 is much weaker, the observation that block 1 is not good is consistent. Results for other environments are in Appendix J.6.1.

### 5.6.2 Attention Distance Analysis

In previous sections, we find that the information improving learning efficiency is preserved in all blocks only in the varying amount and that is probably context-related. To further confirm that the context-like information is indeed preserved, we investigate how the model processes context by calculating the attention distance [59] of each Transformer block for the result of Section 5.5. Attention distance is the average distance between key and query weighted by attention weight. Previous studies successfully used this measure to reveal how locally Transformers process context [43, 59]. Attention distance will elucidate how far away each model processes the tokens of input.

In particular, we compute the gap ($|d_{att}(epoch) - d_{att}(0)|$) between attention distance at the initial state ($d_{att}(0)$) and that at an epoch ($d_{att}(epoch)$) to study to what extent the model preserves the way to utilize context. If the gap is smaller, it means that the attention distance is more preserved. To clarify the difference between the language-pre-trained model and the randomly initialized model, we compare the gap between epoch 0 and 4 since at epoch 4 for all environments the randomly initialized

---

[7]Note that since the medium dataset we use is corrected by a sub-optimal policy, a better action error does not always result in a better return. We explain this more in detail in Appendix O.

model still produces a low return, while the GPT2 achieves a high return (Fig. 8). The detail of the setup and results for other epochs are shown in Appendices I.2.1 and J.6.2.

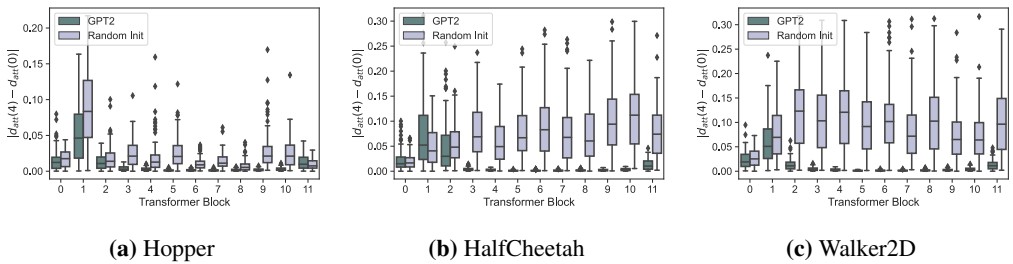

**(a)** Hopper        **(b)** HalfCheetah        **(c)** Walker2D

**Figure 10:** Attention distance gap between epoch 0 and epoch 4.

Results are shown in Fig. 10, where the x-axis is the Transformer block index and the y-axis is the gap between attention distances. Fig. 10 (a), shows that the attention gaps of GPT2 except for block 1 are small, indicating that attention distances are kept from the pre-training in these blocks. Taken together with the finding of Section 5.6.1 that blocks other than block 1 improve learning efficiency, these preserved attention distances are likely to positively affect fine-tuning. This result further supports the hypothesis that language-pre-training brings the model context-like information and the use of this information allows efficient learning.

We also notice the outstanding contrast that gaps of language-pre-trained models are almost zero except for block 1 (and block 2 for HalfCheetah), while those for the randomly initialized model are large. This trend is clearer from middle to deep blocks (3 to 10) for all environments. Turning an eye to Fig. 9 (a), we notice that the action error of these blocks decreases faster than the other block (0 to 2 and 11). Thus, we conclude that the pre-trained context-like information could help the training by enabling the model to predict action more accurately.

We summarize findings of Section 5 with particular attention to the implication for performance:

- 5.1: Re-using representation is not the cause of the good performance of GPT2.
- 5.2: Fitting to data better is not the cause of the good performance of GPT2.
- 5.3: The good performance of GPT2 might come from some unchanged parameters.
- 5.4: Gradient clipping and large gradient might be a cause of the bad performance of iGPT.
- 5.5: GPT2 can learn efficiently even without the context provided.
- 5.6.1: Even a single pre-trained Transformer block makes training more efficient.
- 5.6.2: A cause of the good performance of GPT2 is a pre-acquired way to use context.

# 6 Discussion

**Conclusion** We have examined how pre-training on data of different modalities influences fine-tuning to offline RL. Internal representation analysis shows that pre-trained Transformer largely changes its representation while attaining less knowledge of the downstream task. By the analysis of change in parameters, we find that pre-trained Transformers do not change parameters that much and that the bad performance of the image-pre-trained model might partially come from gradient clipping on the large gradient dominated by a few parameters. Fine-tuning models with no context, we find that the language-pre-trained model can efficiently solve offline RL tasks even without context. Follow-up analysis supports the hypothesis that language pre-training probably gives the Transformer context-like information and the model exploits it to tackle the offline RL tasks.

**Discussion** The finding in Section 5.6.2 of the usefulness of middle blocks is interesting, given that the contextual model's middle layers contain syntactic information [60, 61] and are most transferable [20]. This implies that the syntactic capability of Transformers could be related to the performance on offline RL. The role of layer norm is also noteworthy as we observe that it may characterize the behavior in fine-tuning (Sections 5.1 and 5.4). Studying the pre-training on other RL tasks is also critical to understanding the benefit of pre-training in RL. Many studies of human evolution and

development point to a close relationship between language and behavior [62, 63]. Hence, studying the relationship between these modalities matters to create human-like intelligence that handles language. A more detailed analysis of this relationship is a promising research direction.

**Limitation and Impact**  We consider only a random seed and a few datasets from OpenAI Gym for analysis. Thus, using diverse data is important and checking if the claim holds with the average results with many more seeds is crucial. Also, the validity of the metrics we used should be critically examined since different results could be obtained by using more carefully designed ones. Because this work is fundamental research, no immediate serious negative societal impact is expected.

## Acknowledgments and Disclosure of Funding

We thank DEEPCORE Inc. for providing the computational resource.

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
