# Appendix

## Table of Contents

# A  Fine-Tuning Result for Sanity Check

We compare our performance results to those of previous studies to ensure that they are not far off from the results of the prior study [9], the result of which is shown in Table 1. Following the previous studies [38, 35, 9], we report the normalized score of mean return: $100 \times \frac{\text{score} - \text{random score}}{\text{expert score} - \text{random score}}$, where *random score* is mean return generated by a random policy and *expert score* is generated by a policy trained by Soft Actor-Critic [64]. Mean return is the sum of rewards averaged over trajectory. For further details of the datasets and metrics, please refer to the paper that proposes D4RL [38]. Although the previous work [9] used several techniques to improve the performance, e.g. language model co-training, the extension of positional embedding, and similarity encouragement, we do not employ any of these techniques so that we study the pure effect of pre-training. The training details are described in Appendix B.

The result for the previous work is the average and standard deviation of three random seeds, while our result is those of two random seeds. The aim of this comparison is just to confirm that our result is not too pathological, checking soundness with two seeds would be valid enough. For reference, we also include the results of the Decision Transformer (DT) since the randomly initialized model (Random Init) is a large Decision Transformer. Following the previous study [9], which says that it conducted early stopping, we show the result of the best checkpoint in 40 epoch checkpoints. The result shows that our result is not too far away from the previous work, ensuring that the models of the subject of our analysis are valid enough.

# B  Training Details

Hyperparameters are determined following previous study [9]. The hyperparameters and other setups for fine-tuning the model are summarized in Table 3. The number of layers is 12 for GPT2 and randomly initialized model and 24 for iGPT. The other setup not specified, including $\beta$ of Adam optimizer [65] and model initialization scheme follows Pytorch default ones or the default values of optional arguments in the scripts that our experiment is based on [8]. We do not do any pre-training ourselves, but use publicly available pre-trained models explained in Section 3.

Each model is trained to minimize the mean squared loss between its predicted actions $\hat{a}_t$ and the true actions $a_t$. To get the big picture of how fine-tuning of GPT-based models to offline RL data is conducted, please refer to the pseudo-code of the previous work [35].

The D4RL mujoco task used as offline RL data for our analysis has *expert*, *medium*, *medium-replay*, *medium-expert*, and *random* datasets. The *expert* dataset is the one trained by Soft Actor-Critic [64], *medium* is the one partially trained by Soft Actor-Critic and was stopped early, *medium-replay* is the one accumulated in the replay buffer before the model reached *medium*'s level, and *medium-expert* is a mixture of *medium* and *expert* results. The *random* dataset is the trajectory collected by the random policy. We conducted our experiments on the *medium* data set since it was used by the previous study [9] and our analysis is based on the observations of this previous study. The *medium* dataset we used is 1 million time steps collected by the policy explained above. For the detail of the dataset, please refer to the original paper [38].

---

[8] `experiment.py` in the following repository:
https://github.com/machelreid/can-wikipedia-help-offline-rl/blob/main/code/experiment.py

**Table 3:** Training configuration.

| | |
|---|---|
| # Layers | 12 |
| Emb. Dim. | 768 |
| # Attention Heads | 1 |
| Batch size | 64 |
| Context | 20 |
| Return-to-go conditioning | 6000 HalfCheetah
3600 Hopper
5000 Walker |
| Dropout | 0.2 |
| Learning rate | $1e - 4$ |
| LR Warmup | 5000 steps |
| Epoch | 40 |
| # Steps per Epoch | 2500 |
| Optimizer | Adam |
| Weight Decay | $1e - 4$ |

## C  Model Architecture and Module Names

In the following sections, we use the module names of our models (`gpt2` and `openai/imagegpt-small`) to describe the experimental details. Thus, for the reference, we show the model architecture we use and the names of the modules of the model. We show that of `gpt2` but the configuration is the same for `openai/imagegpt-small` except for the number of Transformer blocks. For the detailed configurations of pre-trained models, please refer to the following links:

- `gpt2`: https://huggingface.co/gpt2
- `openai/imagegpt-small`: https://huggingface.co/openai/imagegpt-small

We show the model architecture below, following the Pytorch format:

```
1  DecisionTransformer(
2    (transformer): GPT2Model(
3      (wte): Embedding(50257, 768)
4      (wpe): Embedding(1024, 768)
5      (drop): Dropout(p=0.1, inplace=False)
6      (h): ModuleList(
7        (0): GPT2Block(
8          (ln_1): LayerNorm((768,), eps=1e-05, elementwise_affine=True)
9          (attn): GPT2Attention(
10           (c_attn): Conv1D()
11           (c_proj): Conv1D()
12           (attn_dropout): Dropout(p=0.1, inplace=False)
13           (resid_dropout): Dropout(p=0.2, inplace=False)
14         )
15         (ln_2): LayerNorm((768,), eps=1e-05, elementwise_affine=True)
16         (mlp): GPT2MLP(
17           (c_fc): Conv1D()
18           (c_proj): Conv1D()
19           (act): NewGELUActivation()
20           (dropout): Dropout(p=0.2, inplace=False)
21         )
22       )
23       (1): GPT2Block(
24         (ln_1): LayerNorm((768,), eps=1e-05, elementwise_affine=True)
```

```
25        (attn): GPT2Attention(
26          (c_attn): Conv1D()
27          (c_proj): Conv1D()
28          (attn_dropout): Dropout(p=0.1, inplace=False)
29          (resid_dropout): Dropout(p=0.2, inplace=False)
30        )
31        (ln_2): LayerNorm((768,), eps=1e-05, elementwise_affine=True)
32        (mlp): GPT2MLP(
33          (c_fc): Conv1D()
34          (c_proj): Conv1D()
35          (act): NewGELUActivation()
36          (dropout): Dropout(p=0.2, inplace=False)
37        )
38      )
39      (2): GPT2Block(
40      .
41      .
42      .
43      )
44      (11): GPT2Block(
45        (ln_1): LayerNorm((768,), eps=1e-05, elementwise_affine=True)
46        (attn): GPT2Attention(
47          (c_attn): Conv1D()
48          (c_proj): Conv1D()
49          (attn_dropout): Dropout(p=0.1, inplace=False)
50          (resid_dropout): Dropout(p=0.2, inplace=False)
51        )
52        (ln_2): LayerNorm((768,), eps=1e-05, elementwise_affine=True)
53        (mlp): GPT2MLP(
54          (c_fc): Conv1D()
55          (c_proj): Conv1D()
56          (act): NewGELUActivation()
57          (dropout): Dropout(p=0.2, inplace=False)
58        )
59      )
60    )
61    (ln_f): LayerNorm((768,), eps=1e-05, elementwise_affine=True)
62  )
63  (embed_timestep): Embedding(1000, 768)
64  (embed_return): Linear(in_features=1, out_features=768, bias=True)
65  (embed_state): Linear(in_features=11, out_features=768, bias=True)
66  (embed_action): Linear(in_features=3, out_features=768, bias=True)
67  (embed_ln): LayerNorm((768,), eps=1e-05, elementwise_affine=True)
68 )
```

**Listing 1:** Model architecture.

Among the above modules, for our analysis, we focus on the modules in `DecisionTransformer.transformer.h`. We summarize the module-related notations used in this paper below for reference:

- *Transformer blocks*: modules directly under `DecisionTransformer.transformer.h`
  - e.g. `DecisionTransformer.transformer.h[0]`.
- *Outputs of blocks*: outputs of `mlp.dropout` of blocks.
  - e.g. `DecisionTransformer.transformer.h[0].mlp.dropout`.
- *Layers*: modules under Transformer blocks.
  - e.g. `DecisionTransformer.transformer.h[0].attn.c_attn`.
- *Parameter set*: parameters of each layer.
  - e.g. `DecisionTransformer.transformer.h[0].ln_1.weight`.

Note that these notations are sometimes used interchangeably as long as it doesn't significantly deteriorate the readability.

# D Activation Similarity

## D.1 Details of Experiments

We randomly sample 100 samples and compute unbiased estimators [39] of linear CKA for these 100 activation vectors. The activation to be analyzed are outputs from all *layers*. In the Decision Transformer, the effective total input length is context length $K$ times the number of token types $(\hat{R}, \boldsymbol{s}, \boldsymbol{a})$. As a result, the shape of the activation obtained at each layer is (`batch size, 3x context length, embedding dimension`). In this study, we compute the CKA for the activity of shape (`batch size, embedding dimension`) that corresponds to the last time step in the context of return-to-go, state, and action, respectively. In other words, noting activation as in the form of python NumPy array `activation`, we obtain `activation[:, -1, :]`, `activation[:, -2, :]`, and `activation[:, -3, :]`. In the previous work analyzing the internal representation of Transformer [41], the dimension of context and representation seem to be concatenated: `activation.contiguous().view(activation.shape[0] -1)`. Since the representation obtained in this way is hard to interpret, we decide to use the way we described above to obtain activation. A descriptive diagram for activation that we compute CKA about is shown in Fig. 11. The design of the diagram is based on a previous study [35].

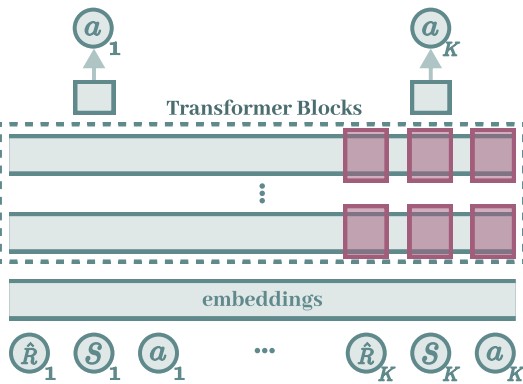

**Figure 11:** Activation we consider to compute CKA.

Each rectangle colored in red is an activation we consider. The left-most rectangles are the activation with return-to-go as input, the middle rectangles are the that with status as input, and the right-most rectangles are that with action as input. Since we consider only the last in the context, the rectangles are drawn only where the tokens of the K-th step are received as input. In Section 5.1, we show the results of activation that $\boldsymbol{s}_K$ is fed in the diagram.

## D.2 Centered Kernel Alignment (CKA)

We show the definition of Centered Kernel Alignment (CKA). Given two activation vectors $\boldsymbol{X} \in \mathbb{R}^{m \times p_1}$ and $\boldsymbol{Y} \in \mathbb{R}^{m \times p_1}$, where $m$ is data size and $p_1$ and $p_2$ are dimensions of hidden layers, the CKA of the two representations is as follows:

$$\text{CKA} = \frac{\text{HSCI}(\boldsymbol{K}, \boldsymbol{L})}{\sqrt{\text{HSCI}(\boldsymbol{K}, \boldsymbol{K})\text{HSCI}(\boldsymbol{L}, \boldsymbol{L})}}, \tag{1}$$

where HSCI is the Hilbert-Schmidt independence criterion [66] and $\boldsymbol{K}_{ij} = k(\boldsymbol{x}_i, \boldsymbol{x}_j)$ and $\boldsymbol{L}_{ij} = l(\boldsymbol{x}_i, \boldsymbol{x}_j)$ are kernels. Particularly, linear CKA is defined as follows:

$$\text{linear CKA} = \frac{||\boldsymbol{Y}^\top \boldsymbol{X}||_F^2}{||\boldsymbol{X}^\top \boldsymbol{X}||_F ||\boldsymbol{Y}^\top \boldsymbol{Y}||_F}, \tag{2}$$

where $|| \cdot ||_F$ is the Frobenius norm. For further details, please refer to the previous work [39].

## D.3 Layer Names Whose CKA is Above a Threshold

In Section 5.1, we say we observe that most of the high CKAs of Fig. 1 (b), which are greater than 0.38, are those of layer normalization. The reason we set this seemingly arbitrary threshold is that we

observe higher layers in a block than other layers in Fig. 1 and want to identify these layers. By eye inspection, we find the threshold over which all these points are included in Fig. 1 is around 0.38 and so we select this value as a threshold. Here, we show the raw result of the observation. The layer name list is shown below:

```
1  0.ln_1, 0.attn.c_attn, 0.ln_2, 0.mlp.c_fc, 0.mlp.c_proj, 0.mlp.dropout
2  1.ln_1, 1.attn.c_attn, 1.ln_2
3  2.ln_1,                2.ln_2
4  3.ln_1,                3.ln_2, 3.mlp.c_fc
5  4.ln_1, 4.attn.c_attn, 4.ln_2
6  5.ln_1, 5.attn.c_attn, 5.ln_2
7  6.ln_1, 6.attn.c_attn, 6.ln_2
8  7.ln_1, 7.ln_2
9  8.ln_1,                8.ln_2
10 9.ln_1, 9.ln_2
11 10.ln_1, 10.attn.c_attn, 10.ln_2
12 11.ln_1.               11.ln_2
```

**Listing 2:** Layer names whose CKA is above a threshold

Note that, for example, `0.ln_1` is the first layer normalization layer (`ln_1`) of the first transformer block (`0`). We observe that 24 of 34 layers are layer norm modules (`ln_1` and `ln_2`).

## D.4   CKA Between Different Models

To check if pre-trained models and the randomly initialized model converge to a similar representation or not, we compute the CKA between representations of fine-tuned these models. The result is shown in Fig. 12, where (a) is the CKA between GPT2 and iGPT, (b) is that between GPT2 and random initialization, and (c) is that between iGPT and random initialization. We observe that CKA values in these heat maps are small, confirming that different models learn different representations.

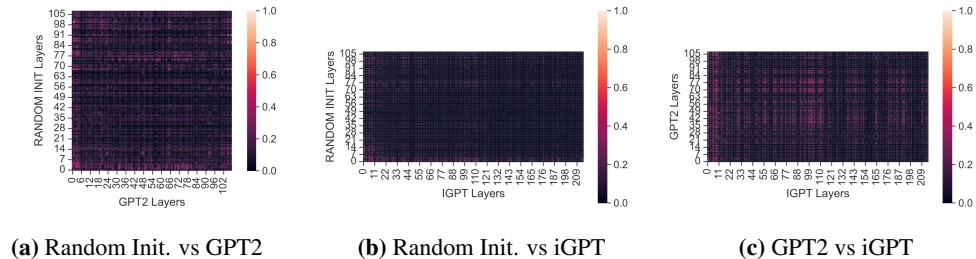

**(a)** Random Init. vs GPT2        **(b)** Random Init. vs iGPT        **(c)** GPT2 vs iGPT

**Figure 12:** CKA similarity across layers between different fine-tuned models.

## D.5   CKA Between Different Layers in a Model

### D.5.1   Post-Fine-Tuning

In Appendix D.4, we find that each layer of the pre-trained model and randomly initialized model learn representation differently. In this section, we delve into what representation these models acquire. For that purpose, we compute CKA values of the different layers in the same model after fine-tuning. The result for the state input token with the *Hopper-medium* dataset is shown in Fig. 13. The results for other environments and input tokens are in Appendix J.1.

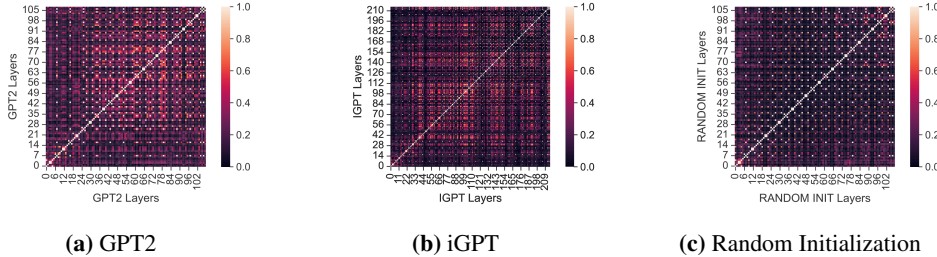

**(a)** GPT2         **(b)** iGPT         **(c)** Random Initialization

**Figure 13:** CKA of different layers in the same models (epoch 40).

Comparing the CKA matrix of pre-trained models (Fig. 13 (a) and Fig. 13 (b)) with that of the randomly initialized model (Fig. 13 (c)), we observe that the randomly initialized model has an almost equally divided lattice structure, while pre-trained models have some block-like structure. In particular, the shallow blocks and the middle to top blocks of language-pre-trained model have a bit similar representation, respectively. This result indicates that probably layers of random initialization process information separately, while some layers of language-pre-trained models coordinate to represent information.

### D.5.2    Pre-Fine-Tuning

For just reference, we put the CKA matrix of pre-trained models before fine-tuning as well. The result is shown in Fig. 14. We observe the grid structure at the initial state as well.

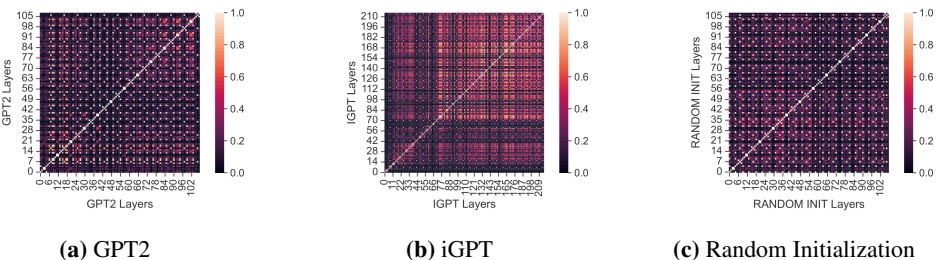

**(a)** GPT2         **(b)** iGPT         **(c)** Random Initialization

**Figure 14:** CKA of different layers in the same models (epoch 0).

## E    Mutual Information Between Hidden Representation and Data

### E.1    Details of Experiments

MINE, the definition of which is explained in Appendix E.2, estimates mutual information by training a neural network. The neural network we use is a feed-forward ReLU network with two hidden layers of width 400. We train the neural network for 1000 iterations by Adam optimizer with a learning rate of $1e-4$. Same as in Appendix D.1, we randomly sample 100 trajectories and use them to obtain the activation. Thus, the dataset for mutual information calculation is the pair between these 100 activation vectors and 100 trajectories for $\hat{I}(X;T)$ and 100 last action vectors for $\hat{I}(Y;T)$. If the estimated value is `NaN`, we exclude the point from the figure.

The descriptive diagram that shows hidden activation and data we consider to compute estimated mutual information is Fig. 15, where (a) is for $\hat{I}(X;T)$ and (b) is for $\hat{I}(Y;T)$. This is an example of mutual information for the deep Transformer block. For the middle block, we use block 6 for GPT2 and random initialization and block 12 for iGPT. Specifically, the outputs of the shallow, middle, and deep Transformer blocks are the outputs of `0.mlp.dropout`, `6.mlp.dropout`, and `11.mlp.dropout` for GPT2 and randomly initialized models and `0.mlp.dropout`, `12.mlp.dropout`, and `23.mlp.dropout` for iGPT.

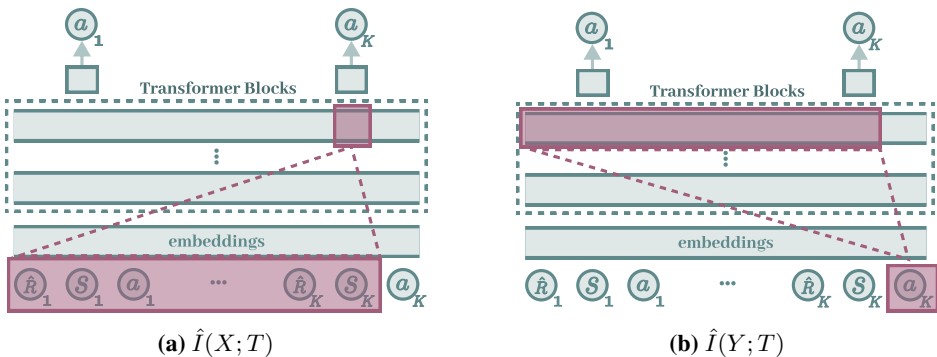

**(a)** $\hat{I}(X;T)$        **(b)** $\hat{I}(Y;T)$

**Figure 15:** Data and activation we consider to compute estimated mutual information.

## E.2 Mutual Information Neural Estimation (MINE)

Given random variables $X$ and $T$, whose joint probability is $\mathbb{P}_{XT}$ and marginal distributions are $\mathbb{P}_X = \int_{\mathcal{T}} \mathrm{d}\mathbb{P}_{XT}$ and $\mathbb{P}_T = \int_{\mathcal{X}} \mathrm{d}\mathbb{P}_{XT}$, the mutual information of $X$ and $T$ are as follows:

$$I(X;T) = \int_{\mathcal{X} \times \mathcal{T}} \log \frac{\mathrm{d}\mathbb{P}_{XT}}{\mathrm{d}\mathbb{P}_X \otimes \mathbb{P}_T} \mathrm{d}\mathbb{P}_{XT}. \tag{3}$$

Denoting neural network with parameters $\theta \in \Theta$ by $\mathrm{NN}_\theta$ and the empirical distribution of $\mathbb{P}$ characterized by $n$ i.i.d. samples by $\hat{\mathbb{P}}^{(n)}$, MINE is defined as follows [57]:

$$\hat{I}(X;T)_n = \sup_{\theta \in \Theta} \mathbb{E}_{\mathbb{P}_{XT}^{(n)}} [\mathrm{NN}_\theta] - \log \left( \mathbb{E}_{\mathbb{P}_X^{(n)} \otimes \hat{\mathbb{P}}_T^{(n)}} \left[ e^{\mathrm{NN}_\theta} \right] \right). \tag{4}$$

Please refer to the previous study [57] for the details.

## E.3 Mutual Information Without Considering Context

As a complementary analysis, we also calculate the estimated mutual information between hidden representation and $s_t$ and $a_t$. In other words, we compute $\hat{I}(s_t; T_l(s_t))$ and $\hat{I}(a_t; T_l(s_t))$ as $\hat{I}(X;T)$ and $\hat{I}(Y;T)$ for all time steps in the context $t = 1, ..., K$. Then, we take an average of them over the context. The descriptive diagram is shown in Fig. 16.

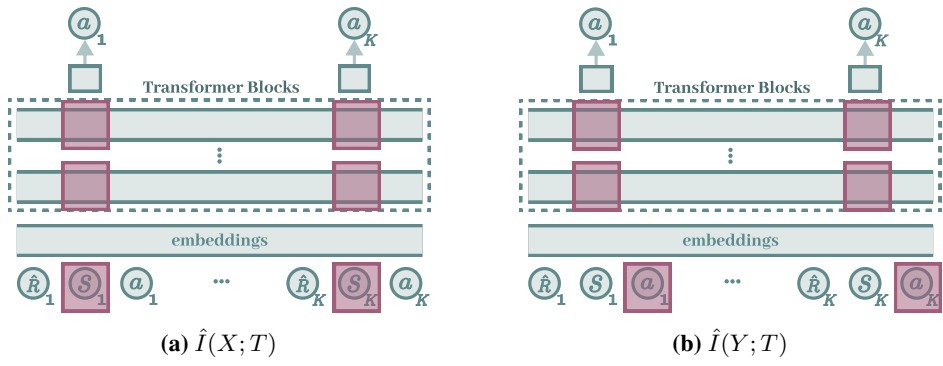

**(a)** $\hat{I}(X;T)$        **(b)** $\hat{I}(Y;T)$

**Figure 16:** Data and activation we consider to compute estimated mutual information.

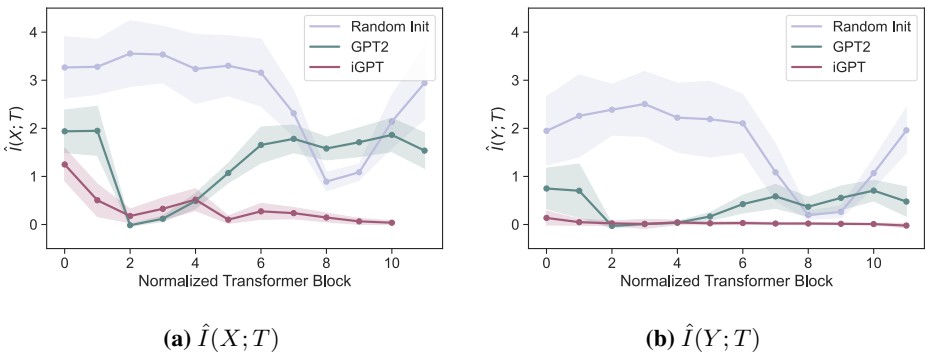

**(a)** $\hat{I}(X;T)$          **(b)** $\hat{I}(Y;T)$

**Figure 17:** Estimated mutual information between hidden representation and state and action.

The result for *Hopper-medium* is shown in Fig. 17. Each point indicates the estimated mean mutual information and the shaded area is the standard deviation. Just as we did in Section 5.1, we take an average over two adjacent elements for iGPT for comparison. We again observe that the randomly initialized model generally has more information on input and label, and iGPT has almost nothing about label-related information.

## F   Parameter Similarity

### F.1   Details of Experiments

The parameters considered in the analysis are those in Transformer blocks (`DecisionTransformer.transformer.h`). We concatenate all these parameters into a single vector and compute $l2$ distance and cosine similarity for this vector. For the post-fine-tuning model in Section 5.3, we use the model trained with the *Hopper-medium* dataset. The results for other environments are in Appendix J.3.

### F.2   Ticks' Labels of Figures 3 and 4

The labels of ticks of Figs. 3 and 4 in Section 5.3 correspond to all *parameter sets*. For example, one of the parameter sets is `0.ln_1.weight`: this is the weight vector of the layer normalization in the first (`0`) transformer block (`h`). The labels for ticks of GPT2 and the randomly initialized model are summarized below:

```
1   '0.ln_1.weight',
2   '0.ln_1.bias',
3   '0.attn.c_attn.weight',
4   '0.attn.c_attn.bias',
5   '0.attn.c_proj.weight',
6   '0.attn.c_proj.bias',
7   '0.ln_2.weight',
8   '0.ln_2.bias',
9   '0.mlp.c_fc.weight',
10  '0.mlp.c_fc.bias',
11  '0.mlp.c_proj.weight',
12  '0.mlp.c_proj.bias',
13  '1.ln_1.weight',
14  '1.ln_1.bias',
15  '1.attn.c_attn.weight',
16  '1.attn.c_attn.bias',
17  '1.attn.c_proj.weight',
18  '1.attn.c_proj.bias',
19  '1.ln_2.weight',
20  '1.ln_2.bias',
21  '1.mlp.c_fc.weight',
22  '1.mlp.c_fc.bias',
23  '1.mlp.c_proj.weight',
24  '1.mlp.c_proj.bias',
```

```
25  '2.ln_1.weight',
26  '2.ln_1.bias',
27  '2.attn.c_attn.weight',
28  '2.attn.c_attn.bias',
29  '2.attn.c_proj.weight',
30  '2.attn.c_proj.bias',
31  '2.ln_2.weight',
32  '2.ln_2.bias',
33  '2.mlp.c_fc.weight',
34  '2.mlp.c_fc.bias',
35  '2.mlp.c_proj.weight',
36  '2.mlp.c_proj.bias',
37  '3.ln_1.weight',
38  '3.ln_1.bias',
39  '3.attn.c_attn.weight',
40  '3.attn.c_attn.bias',
41  '3.attn.c_proj.weight',
42  '3.attn.c_proj.bias',
43  '3.ln_2.weight',
44  '3.ln_2.bias',
45  '3.mlp.c_fc.weight',
46  '3.mlp.c_fc.bias',
47  '3.mlp.c_proj.weight',
48  '3.mlp.c_proj.bias',
49  '4.ln_1.weight',
50  '4.ln_1.bias',
51  '4.attn.c_attn.weight',
52  '4.attn.c_attn.bias',
53  '4.attn.c_proj.weight',
54  '4.attn.c_proj.bias',
55  '4.ln_2.weight',
56  '4.ln_2.bias',
57  '4.mlp.c_fc.weight',
58  '4.mlp.c_fc.bias',
59  '4.mlp.c_proj.weight',
60  '4.mlp.c_proj.bias',
61  '5.ln_1.weight',
62  '5.ln_1.bias',
63  '5.attn.c_attn.weight',
64  '5.attn.c_attn.bias',
65  '5.attn.c_proj.weight',
66  '5.attn.c_proj.bias',
67  '5.ln_2.weight',
68  '5.ln_2.bias',
69  '5.mlp.c_fc.weight',
70  '5.mlp.c_fc.bias',
71  '5.mlp.c_proj.weight',
72  '5.mlp.c_proj.bias',
73  '6.ln_1.weight',
74  '6.ln_1.bias',
75  '6.attn.c_attn.weight',
76  '6.attn.c_attn.bias',
77  '6.attn.c_proj.weight',
78  '6.attn.c_proj.bias',
79  '6.ln_2.weight',
80  '6.ln_2.bias',
81  '6.mlp.c_fc.weight',
82  '6.mlp.c_fc.bias',
83  '6.mlp.c_proj.weight',
84  '6.mlp.c_proj.bias',
85  '7.ln_1.weight',
86  '7.ln_1.bias',
87  '7.attn.c_attn.weight',
88  '7.attn.c_attn.bias',
89  '7.attn.c_proj.weight',
```

```
90   '7.attn.c_proj.bias',
91   '7.ln_2.weight',
92   '7.ln_2.bias',
93   '7.mlp.c_fc.weight',
94   '7.mlp.c_fc.bias',
95   '7.mlp.c_proj.weight',
96   '7.mlp.c_proj.bias',
97   '8.ln_1.weight',
98   '8.ln_1.bias',
99   '8.attn.c_attn.weight',
100  '8.attn.c_attn.bias',
101  '8.attn.c_proj.weight',
102  '8.attn.c_proj.bias',
103  '8.ln_2.weight',
104  '8.ln_2.bias',
105  '8.mlp.c_fc.weight',
106  '8.mlp.c_fc.bias',
107  '8.mlp.c_proj.weight',
108  '8.mlp.c_proj.bias',
109  '9.ln_1.weight',
110  '9.ln_1.bias',
111  '9.attn.c_attn.weight',
112  '9.attn.c_attn.bias',
113  '9.attn.c_proj.weight',
114  '9.attn.c_proj.bias',
115  '9.ln_2.weight',
116  '9.ln_2.bias',
117  '9.mlp.c_fc.weight',
118  '9.mlp.c_fc.bias',
119  '9.mlp.c_proj.weight',
120  '9.mlp.c_proj.bias',
121  '10.ln_1.weight',
122  '10.ln_1.bias',
123  '10.attn.c_attn.weight',
124  '10.attn.c_attn.bias',
125  '10.attn.c_proj.weight',
126  '10.attn.c_proj.bias',
127  '10.ln_2.weight',
128  '10.ln_2.bias',
129  '10.mlp.c_fc.weight',
130  '10.mlp.c_fc.bias',
131  '10.mlp.c_proj.weight',
132  '10.mlp.c_proj.bias',
133  '11.ln_1.weight',
134  '11.ln_1.bias',
135  '11.attn.c_attn.weight',
136  '11.attn.c_attn.bias',
137  '11.attn.c_proj.weight',
138  '11.attn.c_proj.bias',
139  '11.ln_2.weight',
140  '11.ln_2.bias',
141  '11.mlp.c_fc.weight',
142  '11.mlp.c_fc.bias',
143  '11.mlp.c_proj.weight',
144  '11.mlp.c_proj.bias'
```

**Listing 3:** Label list of ticks.

For iGPT, the number of transformer block is twice (from 0-11 to 0-23) but the configuration is the same.

## G  Gradient Analysis

### G.1  Details of Experiments

We randomly sample 100 samples for gradient norm in Fig. 6 and 50 samples for gradient confusion in Fig. 5 and compute the gradient of the loss on each of these samples for all parameters of the models fine-tuned after 1 epoch. The bar plot of Fig. 5 is the minimum of the $50 \times 50 = 2500$ cosine similarities. Each point of Fig. 6 corresponds to the gradient norm of each data sample in 100 gradient norms. The top of the box is the 1st quartile and the bottom of the box is the 3rd quartile of the points. The whiskers extend from the box by 1.5 times the inter-quartile range. For gradient clipping, we use a method in Pytorch [67] and set the maximum norm to be $0.25$, following the previous work [9].

The process of creating Fig. 7 is as follows. We first compute gradient norms per parameter set and examine the bar plot that we will explain in Appendix G.2. Noticing that two peaks exist in the figure, we check the label of the parameter sets and find that they are `0.ln_1.weight` and `0.ln_1.bias`. Thus, we lump the remaining as `others` and only show the ratio in the main body of the paper to tell readers the main finding from the observation in an easy-to-understand manner. The parameter sets considered to compute the parameter set-wise gradient norms are the same as those in Appendix F.2.

### G.2  Gradient Norm for Each Parameter

For visibility, we show only the ratio in Section 5.4 as Fig. 7. The full results for gradient norms of each parameter are shown in Fig. 18. The labels of the ticks on the x-axis are the same as that in Appendix F.2. We immediately notice that the first two parameters (`0.ln_1.weight` and `0.ln_1.bias`) are much larger than the remains.

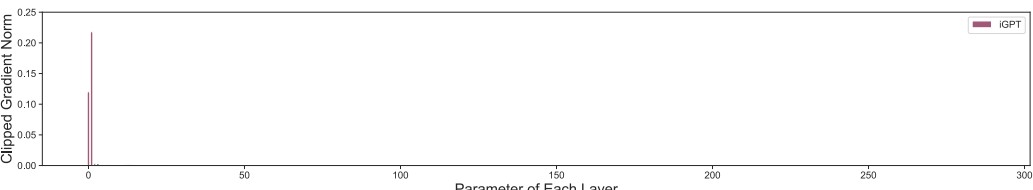

**Figure 18:** Gradient norm of iGPT's each parameter at epoch 1.

### G.3  Analysis of the Effect of Gradient Clipping

In Section 5.4, we noted that large gradient and gradient clipping could be a cause of the bad performance of iGPT. To further explore this point, we conducted a simple experiment.

The experiments of the previous study [9] and our experiments both used a Pytorch function called `torch.nn.utils.clip_grad_norm_` [9] for gradient clipping. This function divides each gradient by the total norm of all gradients and multiplies the clipping constant (gradient norm clipping). However, as we pointed out in Section 5.4, the gradient norm values are dominated by only a few parameters (the layer normalization layer in the first block). This large norm affects the normalization of all gradients, decreasing the informational value of most gradients. Therefore, we hypothesized that one of the possible reasons why iGPT is difficult to learn can be the use of `torch.nn.utils.clip_grad_norm_ function`.

Thus we trained the image-pre-trained model without using gradient clipping. The basic experimental setting is the same as that of Section 5.4 except that we trained the model for 10 epochs, instead of 40 epochs. The results are shown in Figs 19 and 20 for action error and mean return, respectively. The x-axis is the epoch. These figures show that eliminating gradient clipping does not immediately solve the catastrophic performance at least up to 10 epochs. However, we did find that action error is much smaller for no gradient clipping condition, indicating that the learning process of it seems to be

---

[9]Documentation of `torch.nn.utils.clip_grad_norm_` :
https://pytorch.org/docs/stable/generated/torch.nn.utils.clip_grad_norm_.html

more stable and efficient than the clipping was applied. We also found that the return means seem to improve, albeit very slightly. Thus, we can conclude that gradient clipping is a cause of the bad performance of iGPT, though it does not seem to be critical. Exploring the critical factor that makes iGPT performance catastrophic is left a future work.

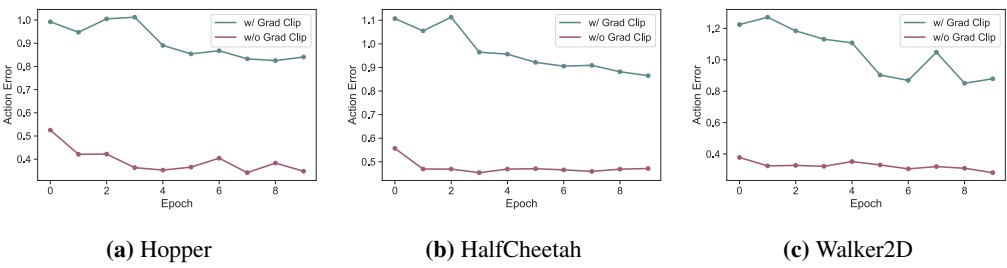

(a) Hopper        (b) HalfCheetah        (c) Walker2D

**Figure 19:** Action error: with gradient clipping v.s. without gradient clipping

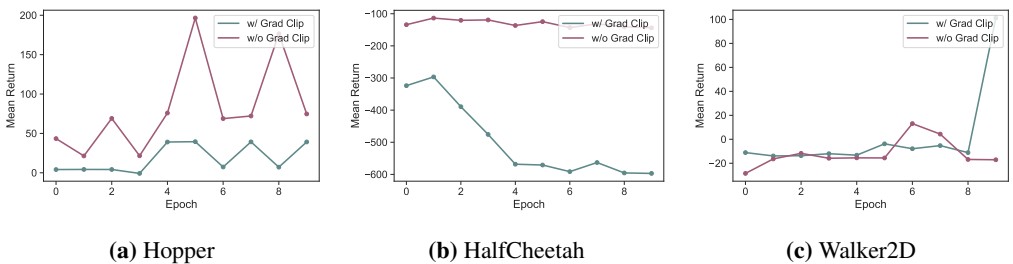

(a) Hopper        (b) HalfCheetah        (c) Walker2D

**Figure 20:** Mean return: with gradient clipping v.s. without gradient clipping

# H    Fine-Tuning with No Context Information

## H.1    Details of Experiments

The configuration of the training is the same as that described in Appendix B except that the context is 1 (no context). We evaluate the model per epoch and the normalized score in Table 4 is the result of the best checkpoint in the 40 checkpoints. The mean return reported in Fig. 8 and Table 4 is the average of the returns over the trajectory. Table 4 is the result of one random seed (seed = 666). The mean and standard deviation of the normalized score for the two seeds (seed = 666 & 42) and the mean return of another seed (seed = 42) are reported in Appendix J.5.

In this paper, we mean *context* by the number of accessible time steps for a model to predict, following the previous study [35]. If the context length is $K$, the model can use inputs in past $K$ steps to predict the action at the current step, while the context length is 1 (no context), the model has to predict only from the current time step.

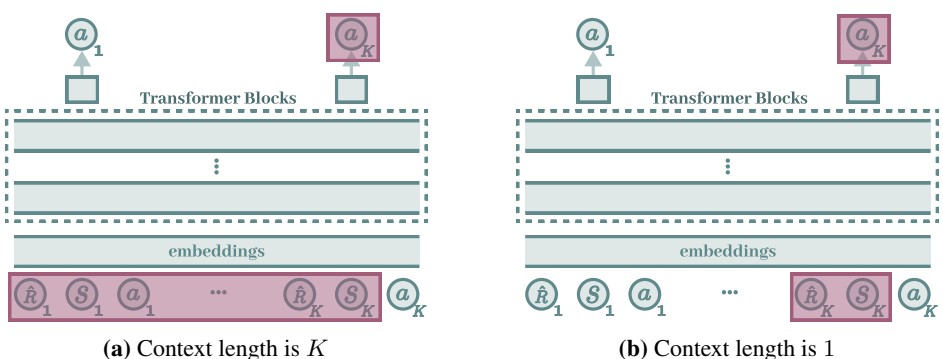

(a) Context length is $K$        (b) Context length is 1

**Figure 21:** Descriptive diagram to explain what we mean by *context*.

## H.2 Analysis of Why Randomly Initialized Model Fails for Hopper with No Context

In Section 5.5, we observe that the performance of randomly initialized models for the Hopper environment is particularly worse than that for other environments (Table 4). In this Section, we further explore a possible cause of this observation.

We speculate that this may be because *how much of the range of context that needs to be looked at* changes depending on the data set. For example, prior studies on decision transformer reported that the effect of context varied depending on the task [35]. Improvement by context means that the context is important information for solving the task. Thus, it could be possible that Hopper is the dataset that requires more information from context than the other two data sets.

As a test of the hypothesis, we randomly sampled a batch sample and calculated the mutual information between action and state or return-to-go at the same time step and compare them between different environments; note that this is the mutual information between the data, not between the data and representation. The higher the value of the mutual information, the higher the mutual dependence between state or return-to-go and action at the same time step. Hence, higher mutual information suggests that the model could predict action better only from the information at the current time step.

In particular, we sample 100 samples of context length $K = 20$. Then, for all time step $t$, we compute the estimated mutual information $\hat{I}(s_t; a_t)$ between state $s_t$ and action $a_t$ and that $\hat{I}(\hat{R}_t; a_t)$ between return-to-go $\hat{R}_t$ and action. Other setup for mutual information estimation is the same as that of Section 5.2. The result is shown in Fig 22, where (a) is the box plot for the state and action and (b) is that for the return-to-go and action. The top of the box is the 1st quartile and the bottom of the box is the 3rd quartile of the points. The whiskers extend from the box by 1.5 times the inter-quartile range. Each point corresponds to different time step in the context.

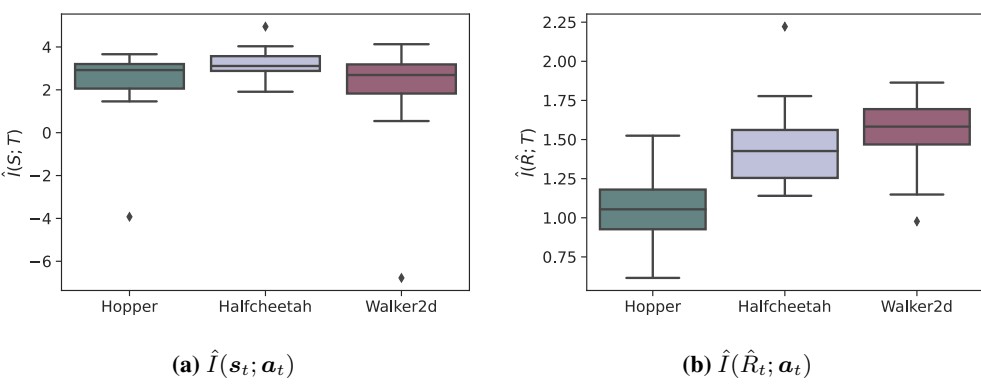

**(a)** $\hat{I}(s_t; a_t)$  **(b)** $\hat{I}(\hat{R}_t; a_t)$

**Figure 22:** Mutual information between data at the same time step in trajectories

We observe that the estimated mutual information between return-to-go and action is smaller on average for Hopper than for other environments (Fig. 22 (b)), though that between action and state does not differ that much (Fig. 22 (a)). Prior research has indicated that return-to-go information seems to be important for prediction [9]. Hence, we can say that models have to use more information from the other steps in the context to solve the Hopper task than models do for other environments. This result supports our hypothesis above.

## I  More In-Depth Analysis of Context Dependence

### I.1  Replacement by the Pre-Trained Block

#### I.1.1  Details of Experiments

The training configuration is the same as that of Section 5.5. The smoothing factor of the exponential moving average for Fig. 9 is 0.8. The action error is the mean square loss between the true action $a_t$ and the predicted action $\hat{a}_t$. Considering that the goal of this experiment is to highlight the effect of pre-training, we train the model for 10 epochs for Hopper and 5 epochs for HalfCheetah and

Walker2D because Fig. 8 shows that the difference between the randomly initialized and pre-trained model is evident during these epochs.

## I.2 Attention Distance Analysis

### I.2.1 Details of Experiments

In the main body of this paper, we mean *attention distance* by *average/mean attention distance* in the previous work [59, 43]. Because we use only one attention head in this study, there is a single point per sample in Fig. 10. We randomly sample 100 trajectory samples and compute attention for these trajectories. When calculating attention distance, we allow the model to access the context of $K = 20$ length and compute the attention distance up to the length. Each point of the box plot (Fig. 10) corresponds to the attention distance of each sample. The configuration for the box plot is the same as that in Appendix G.

# J Results for Other Conditions

## J.1 Activation Similarity

### J.1.1 CKA Between Pre and Post-Fine-Tuning

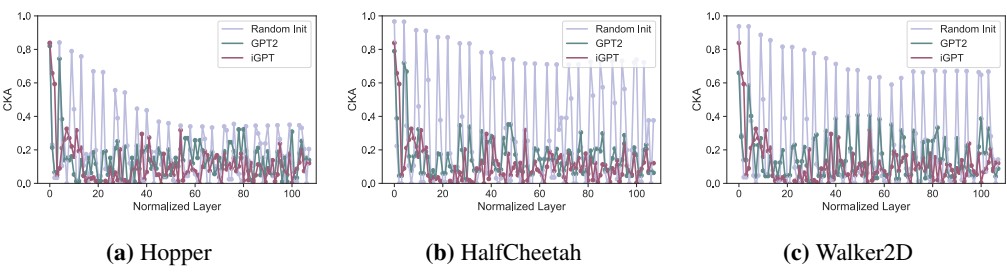

**(a)** Hopper      **(b)** HalfCheetah      **(c)** Walker2D

**Figure 23:** CKA similarity of each layer between pre and post-fine-tuning (Action).

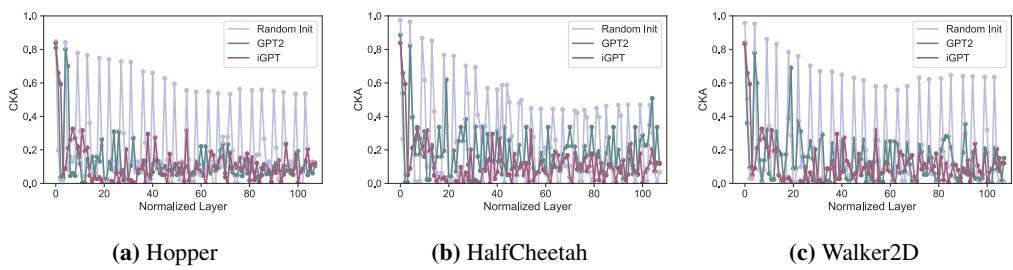

**(a)** Hopper      **(b)** HalfCheetah      **(c)** Walker2D

**Figure 24:** CKA similarity of each layer between pre and post-fine-tuning (Return-to-go).

### J.1.2 CKA Between Pre and Post-Fine-Tuning (Seed = 42)

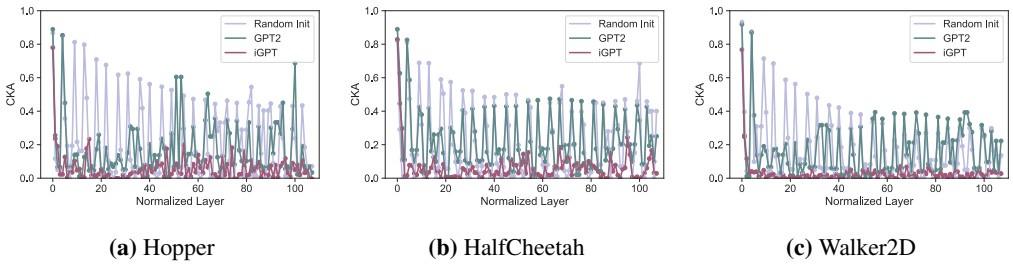

**(a)** Hopper      **(b)** HalfCheetah      **(c)** Walker2D

**Figure 25:** CKA similarity of each layer between pre and post-fine-tuning (State, Seed = 42).

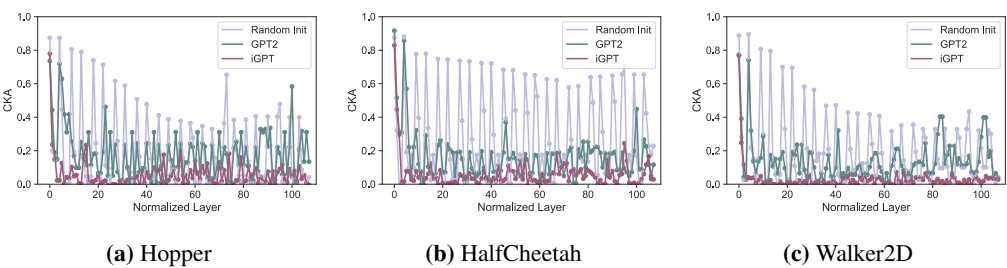

**(a)** Hopper      **(b)** HalfCheetah      **(c)** Walker2D

**Figure 26:** CKA similarity of each layer between pre and post-fine-tuning (Action, Seed = 42).

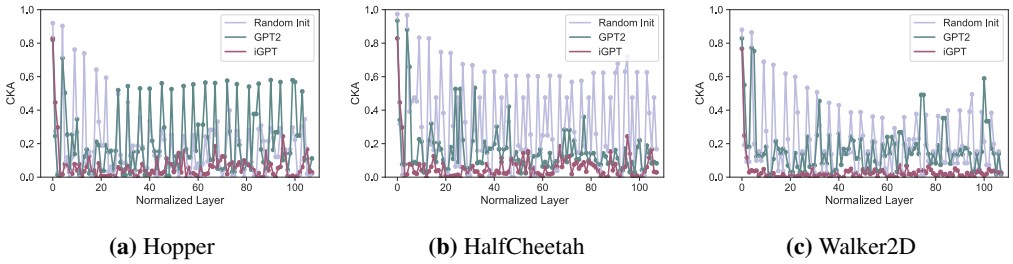

**(a)** Hopper        **(b)** HalfCheetah        **(c)** Walker2D

**Figure 27:** CKA similarity of each layer between pre and post-fine-tuning (Return-to-go, Seed = 42).

### J.1.3 CKA Between Different Models

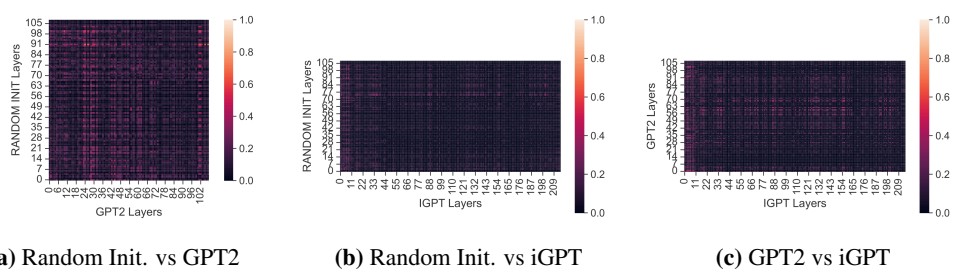

**(a)** Random Init. vs GPT2      **(b)** Random Init. vs iGPT      **(c)** GPT2 vs iGPT

**Figure 28:** CKA between different models (Hopper & Return-to-go).

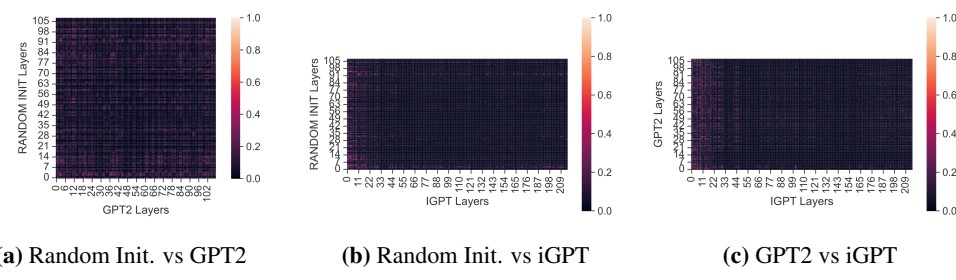

**(a)** Random Init. vs GPT2      **(b)** Random Init. vs iGPT      **(c)** GPT2 vs iGPT

**Figure 29:** CKA between different models (Hopper & Action).

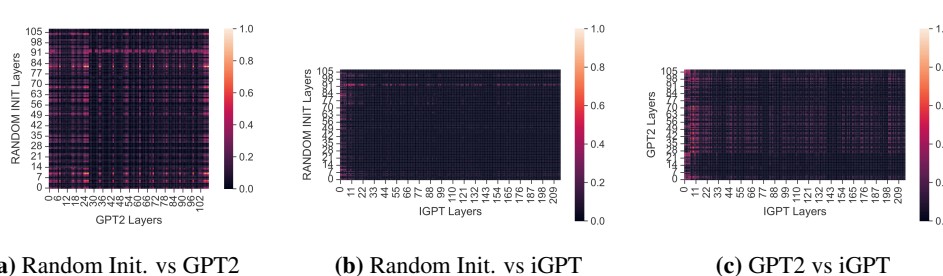

**(a)** Random Init. vs GPT2      **(b)** Random Init. vs iGPT      **(c)** GPT2 vs iGPT

**Figure 30:** CKA between different models (HalfCheetah & State).

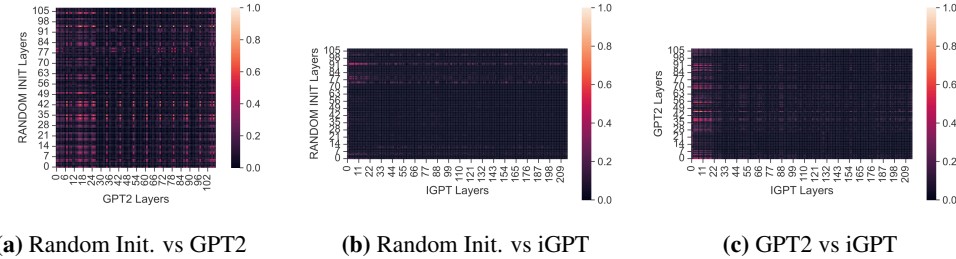

**(a)** Random Init. vs GPT2      **(b)** Random Init. vs iGPT      **(c)** GPT2 vs iGPT

**Figure 31:** CKA between different models (HalfCheetah & Return-to-go).

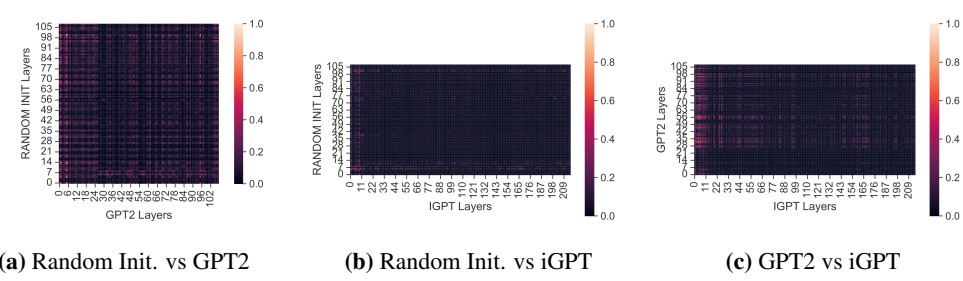

**(a)** Random Init. vs GPT2      **(b)** Random Init. vs iGPT      **(c)** GPT2 vs iGPT

**Figure 32:** CKA between different models (HalfCheetah & Action).

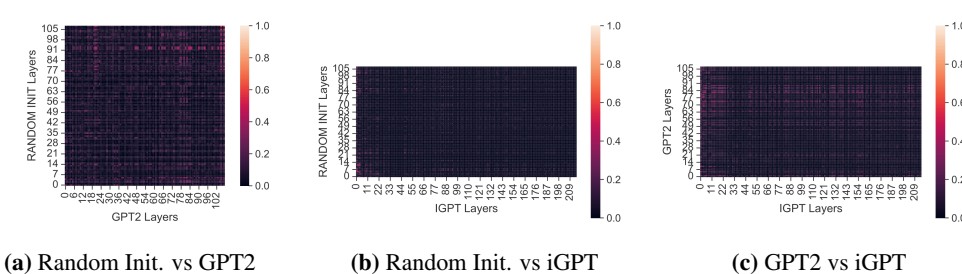

**(a)** Random Init. vs GPT2      **(b)** Random Init. vs iGPT      **(c)** GPT2 vs iGPT

**Figure 33:** CKA between different models (Walker2D & State).

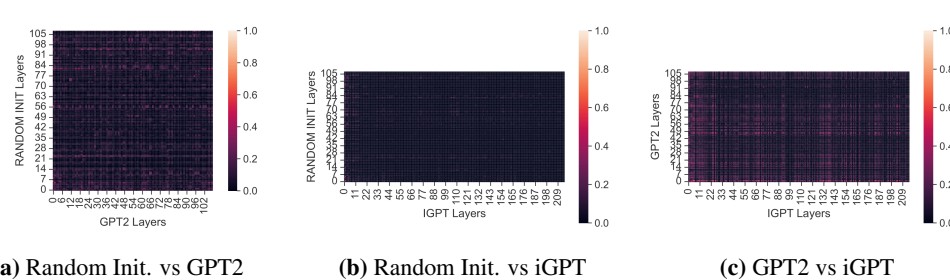

**(a)** Random Init. vs GPT2      **(b)** Random Init. vs iGPT      **(c)** GPT2 vs iGPT

**Figure 34:** CKA between different models (Walker2D & Return-to-go).

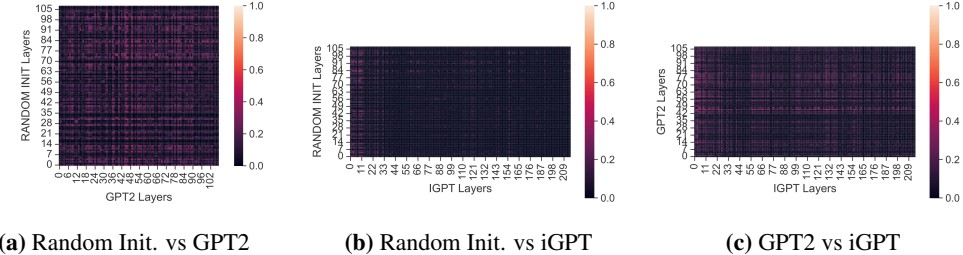

**(a)** Random Init. vs GPT2        **(b)** Random Init. vs iGPT        **(c)** GPT2 vs iGPT

**Figure 35:** CKA between different models (Walker2D & Action).

### J.1.4    CKA Between Different Layers in a Model

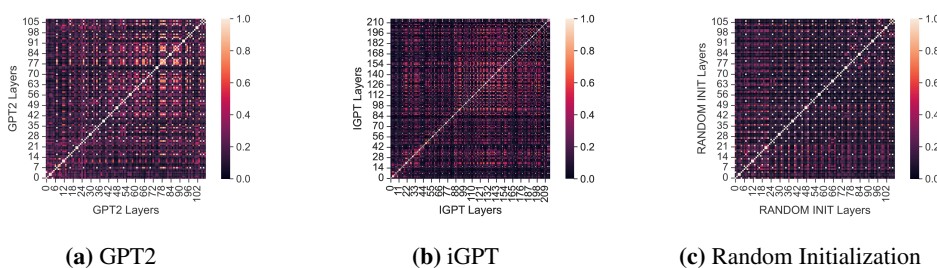

**(a)** GPT2            **(b)** iGPT           **(c)** Random Initialization

**Figure 36:** CKA of different layers in the same model (Hopper & Return-to-go).

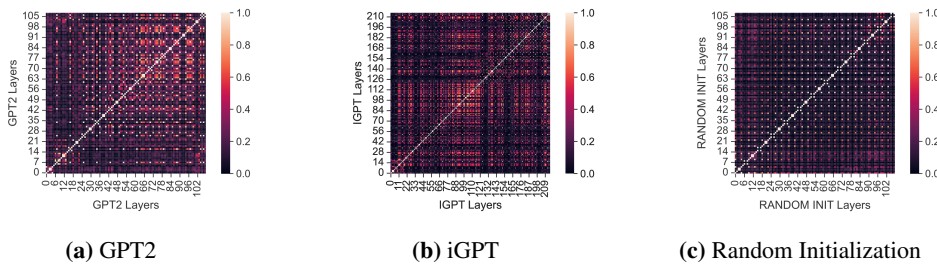

**(a)** GPT2            **(b)** iGPT           **(c)** Random Initialization

**Figure 37:** CKA of different layers in the same model (Hopper & Action).

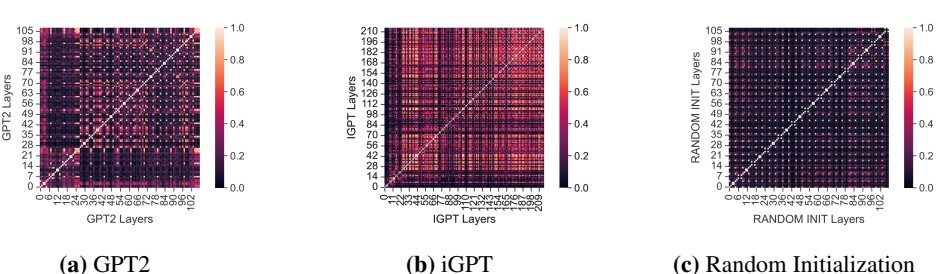

**(a)** GPT2            **(b)** iGPT           **(c)** Random Initialization

**Figure 38:** CKA of different layers in the same model (HalfCheetah & State).

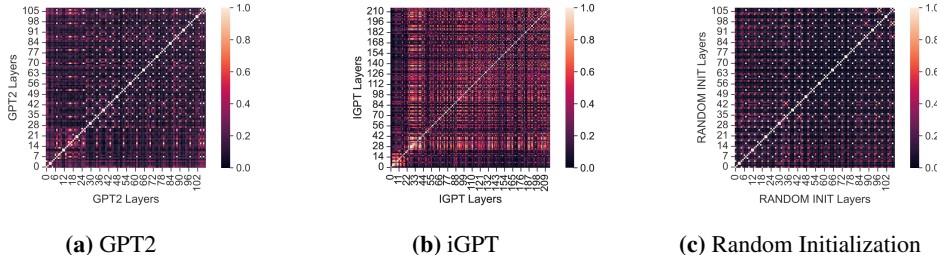

**(a)** GPT2        **(b)** iGPT        **(c)** Random Initialization

**Figure 39:** CKA of different layers in the same model (HalfCheetah & Return-to-go).

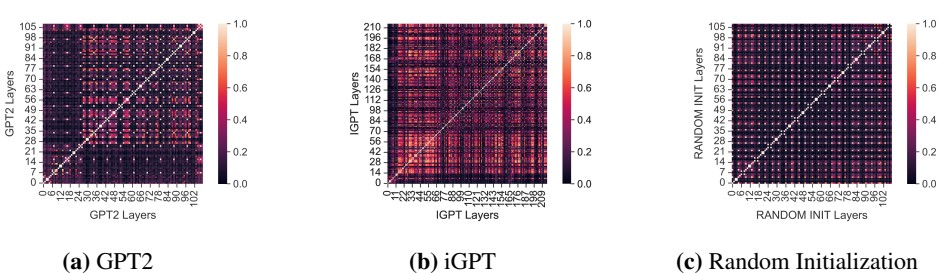

**(a)** GPT2        **(b)** iGPT        **(c)** Random Initialization

**Figure 40:** CKA of different layers in the same model (HalfCheetah & Action).

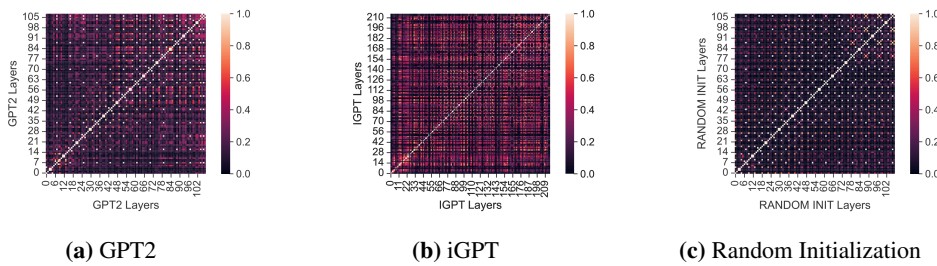

**(a)** GPT2        **(b)** iGPT        **(c)** Random Initialization

**Figure 41:** CKA of different layers in the same model (Walker2D & State).

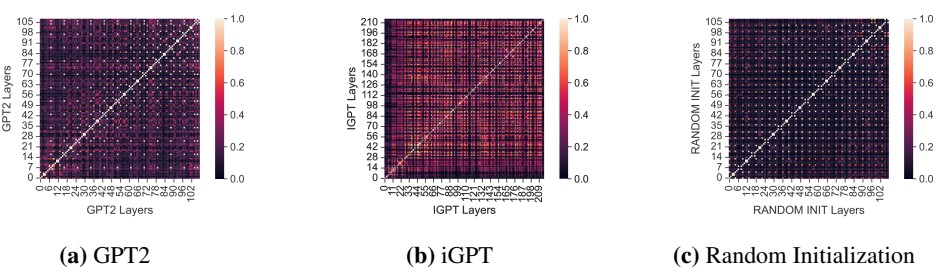

**(a)** GPT2        **(b)** iGPT        **(c)** Random Initialization

**Figure 42:** CKA of different layers in the same model (Walker2D & Return-to-go).

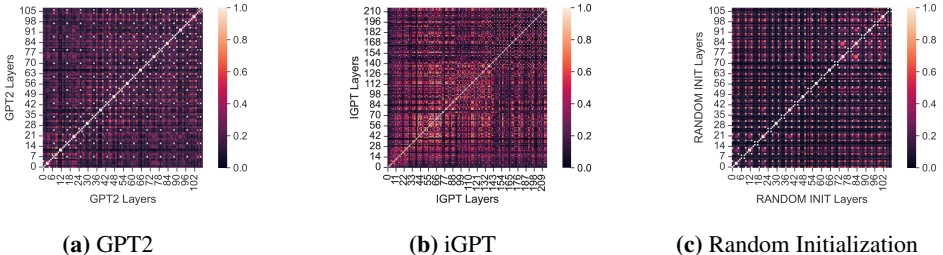

(a) GPT2        (b) iGPT        (c) Random Initialization

**Figure 43:** CKA of different layers in the same model (Walker2D & Action).

## J.2  Mutual Information Between Hidden Representation and Data

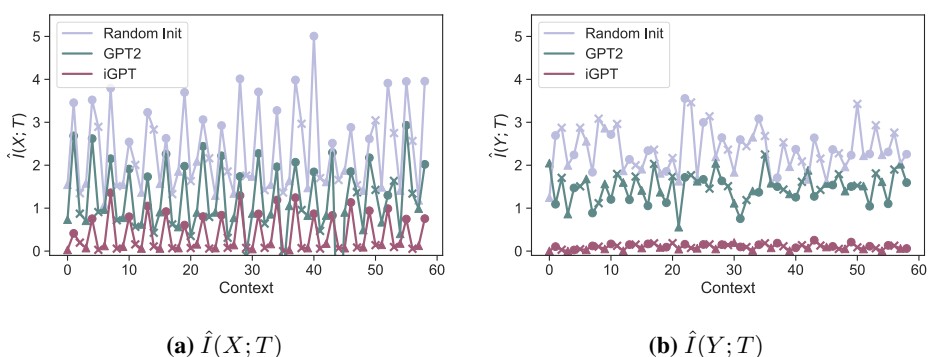

(a) $\hat{I}(X;T)$        (b) $\hat{I}(Y;T)$

**Figure 44:** Estimated mutual information between data and hidden representation (Shallow).

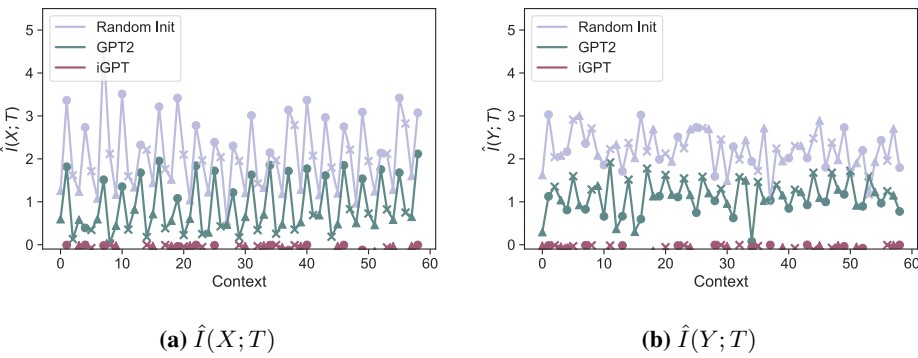

(a) $\hat{I}(X;T)$        (b) $\hat{I}(Y;T)$

**Figure 45:** Estimated mutual information between data and hidden representation (Deep).

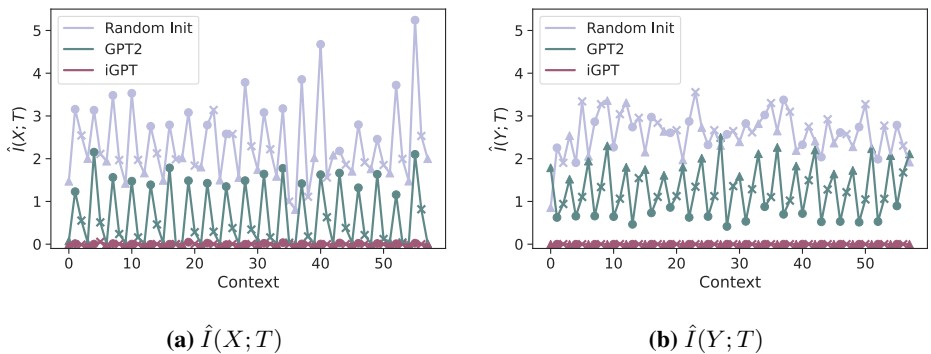

**(a)** $\hat{I}(X;T)$       **(b)** $\hat{I}(Y;T)$

**Figure 46:** Estimated mutual information between data and hidden representation (Middle, Seed = 42).

## J.3 Parameter Similarity

### J.3.1 Parameter Similarity (Other Environment)

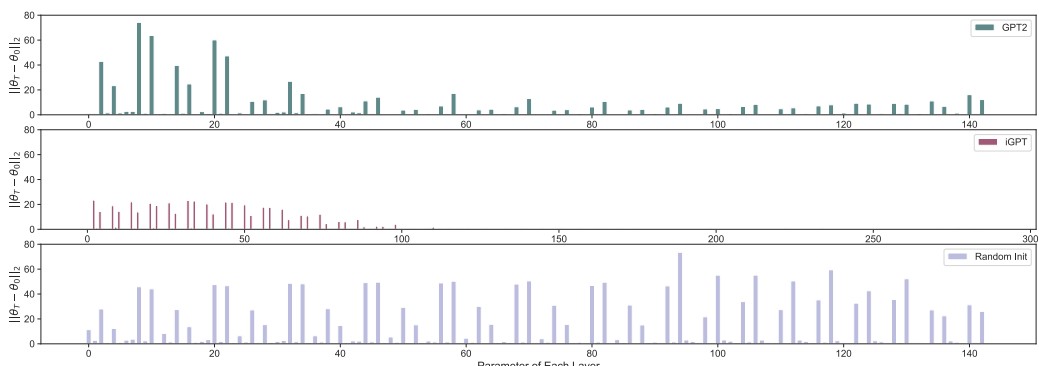

**Figure 47:** L2 distance of each parameter between pre post-fine-tuning (HalfCheetah).

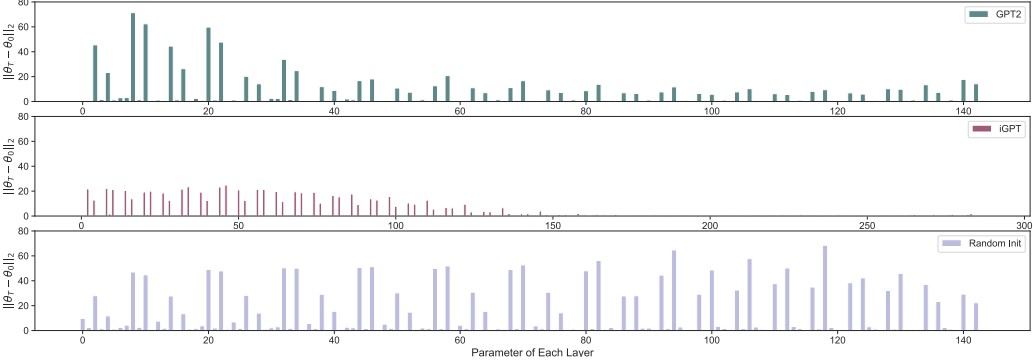

**Figure 48:** L2 distance of each parameter between pre post-fine-tuning (Walker2D).

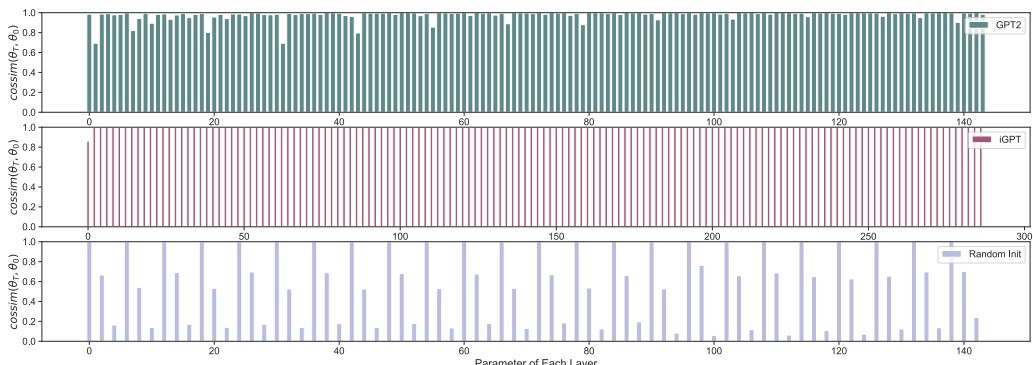

**Figure 49:** Cosine similarity of each parameter between pre post-fine-tuning (HalfCheetah).

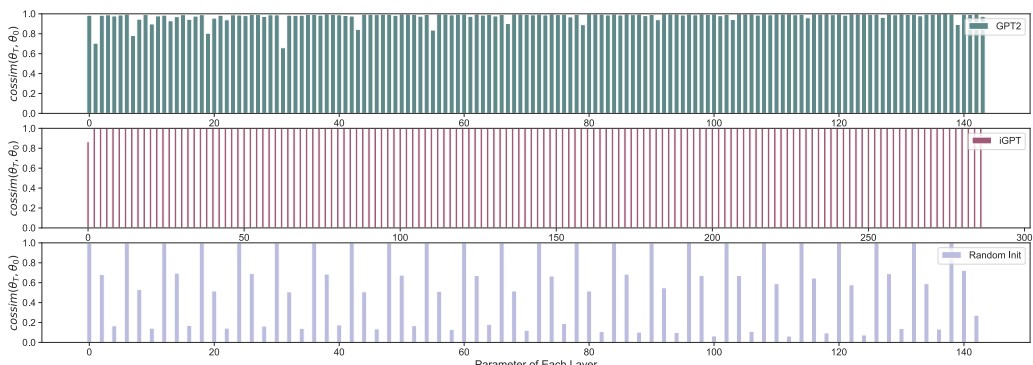

**Figure 50:** Cosine similarity of each parameter between pre post-fine-tuning (Walker2D).

### J.3.2 Parameter Similarity (Seed = 42)

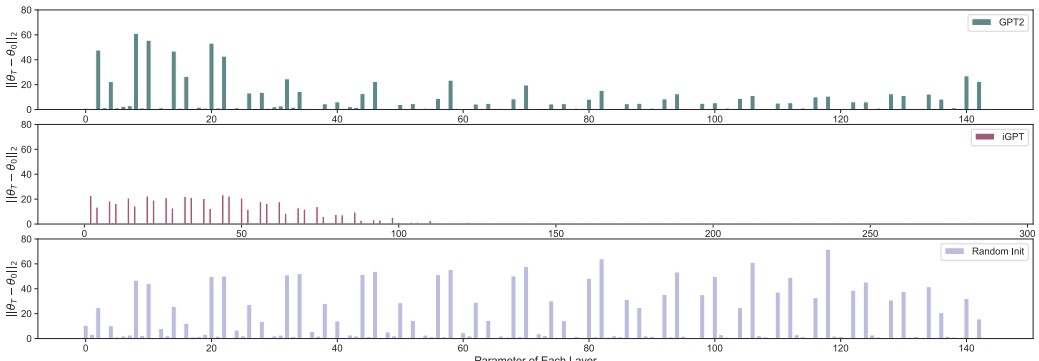

**Figure 51:** L2 distance of each parameter between pre post-fine-tuning (Hopper, Seed = 42).

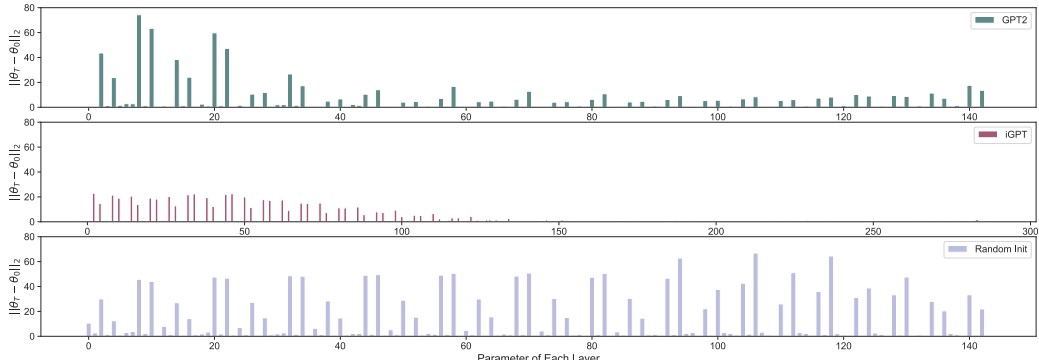

**Figure 52:** L2 distance of each parameter between pre post-fine-tuning (HalfCheetah, Seed = 42).

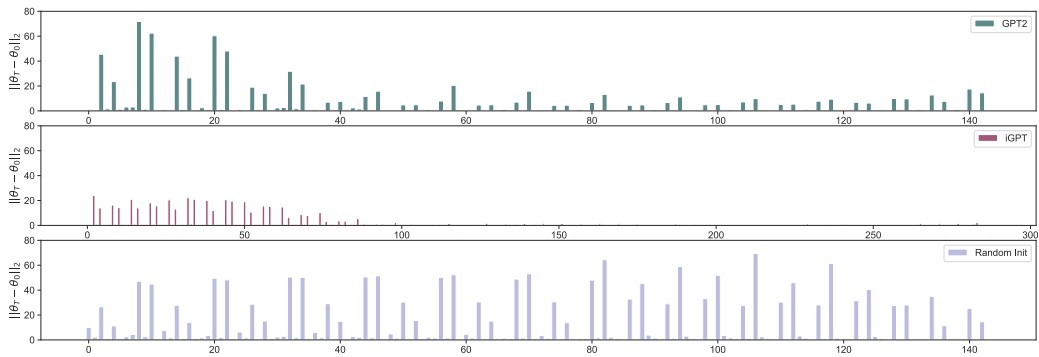

**Figure 53:** L2 distance of each parameter between pre post-fine-tuning (Walker2D, Seed = 42).

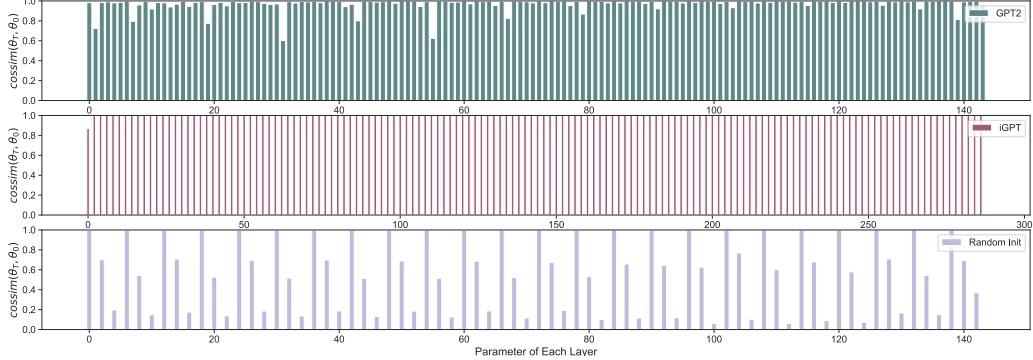

**Figure 54:** Cosine similarity of each parameter between pre post-fine-tuning (Hopper, Seed = 42).

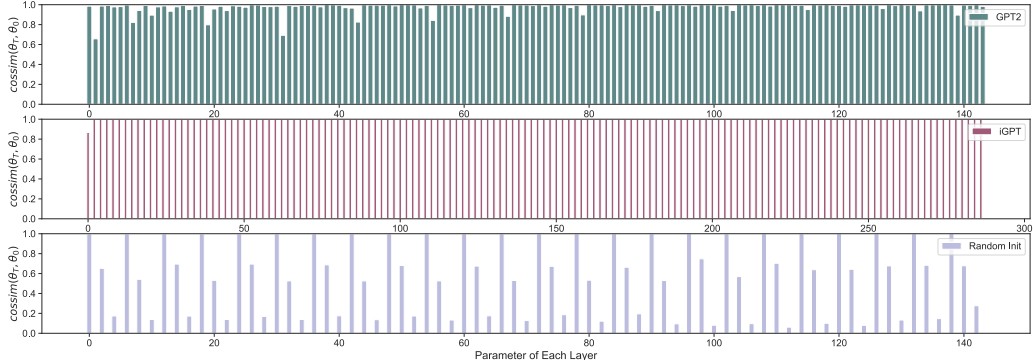

**Figure 55:** Cosine similarity of each parameter between pre post-fine-tuning (HalfCheetah, Seed = 42).

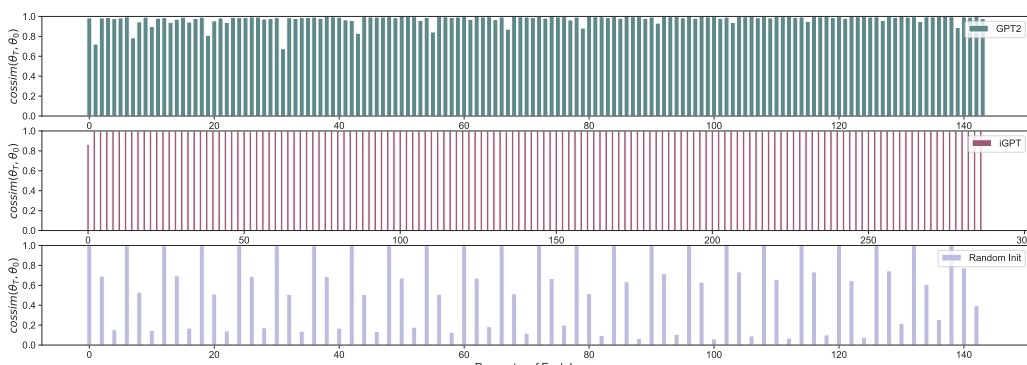

**Figure 56:** Cosine similarity of each parameter between pre post-fine-tuning (Walker2D, Seed = 42).

## J.4 Gradient Analysis

### J.4.1 Gradient Analysis (Other Environments)

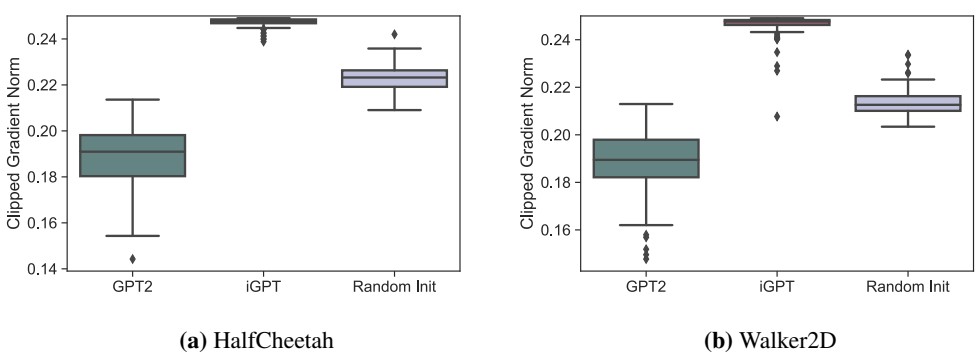

**(a)** HalfCheetah        **(b)** Walker2D

**Figure 57:** Gradient norm.

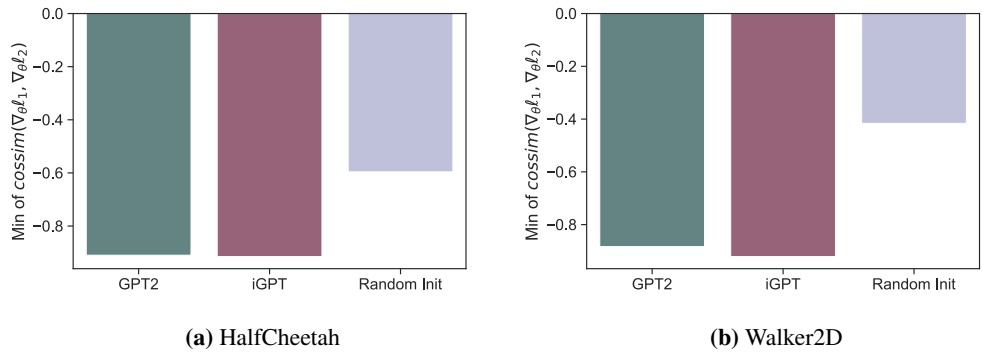

**(a)** HalfCheetah                       **(b)** Walker2D

**Figure 58:** Minimum gradient cosine similarity.

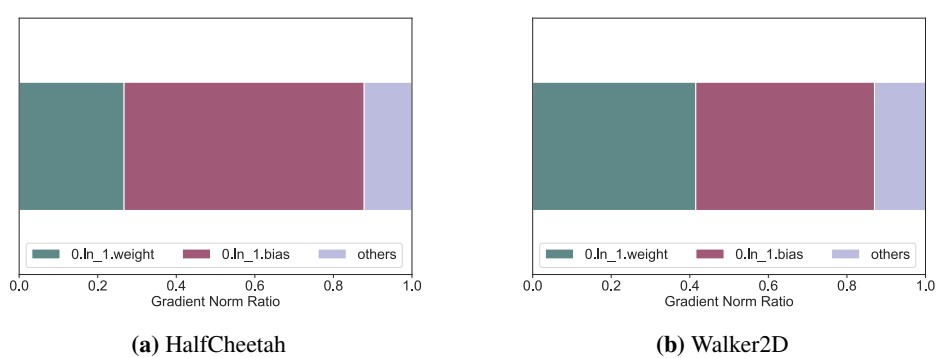

**(a)** HalfCheetah                       **(b)** Walker2D

**Figure 59:** iGPT's gradient norm ratio.

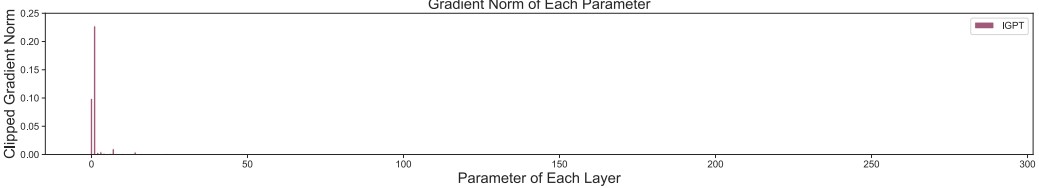

**Figure 60:** Gradient norm of iGPT's each parameter at epoch 1. (HalfCheetah)

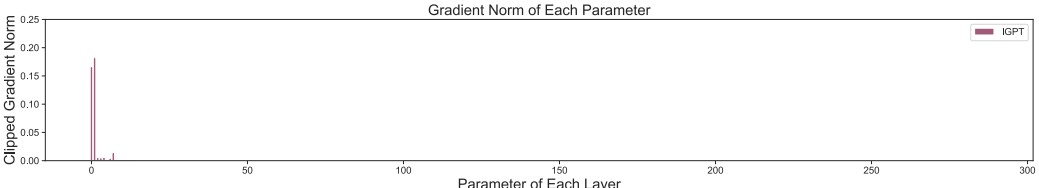

**Figure 61:** Gradient norm of iGPT's each parameter at epoch 1. (Walker2D)

## J.4.2 Gradient Analysis (Seed = 42)

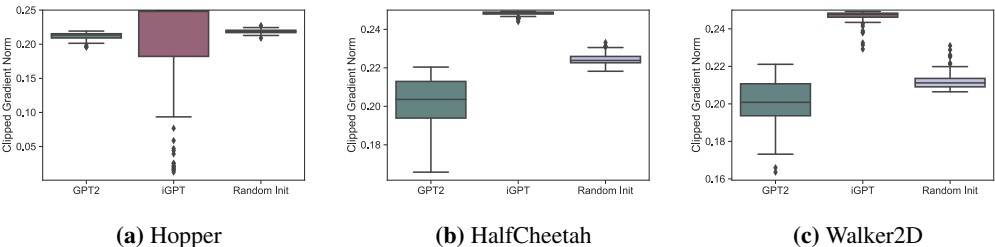

**(a)** Hopper      **(b)** HalfCheetah      **(c)** Walker2D

**Figure 62:** Gradient norm (Seed = 42).

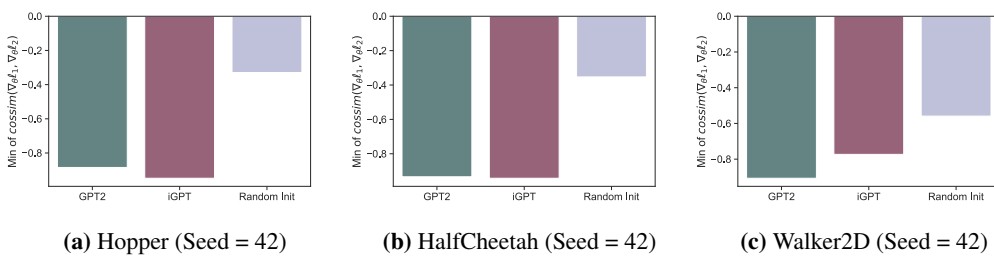

**(a)** Hopper (Seed = 42)      **(b)** HalfCheetah (Seed = 42)      **(c)** Walker2D (Seed = 42)

**Figure 63:** Minimum gradient cosine similarity (seed = 42).

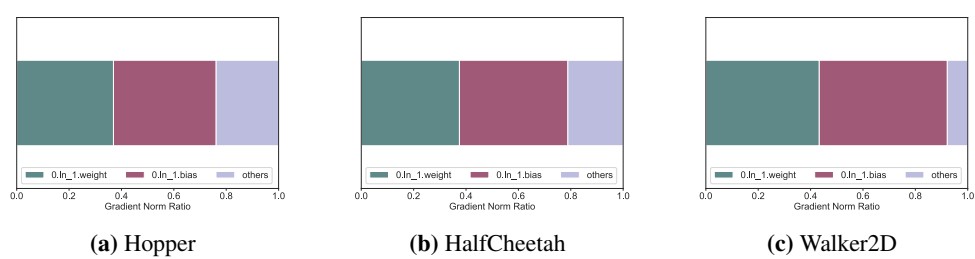

**(a)** Hopper      **(b)** HalfCheetah      **(c)** Walker2D

**Figure 64:** iGPT's gradient norm ratio. (Seed = 42)

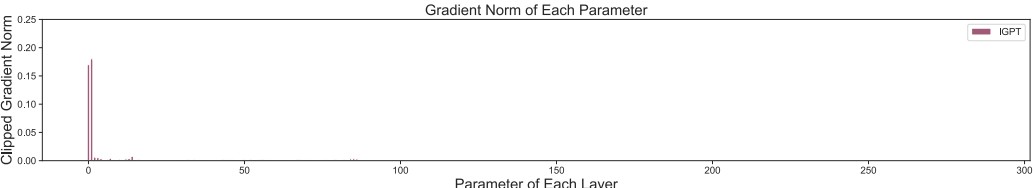

**Figure 65:** Gradient norm of iGPT's each parameter at epoch 1. (Hopper, Seed = 42)

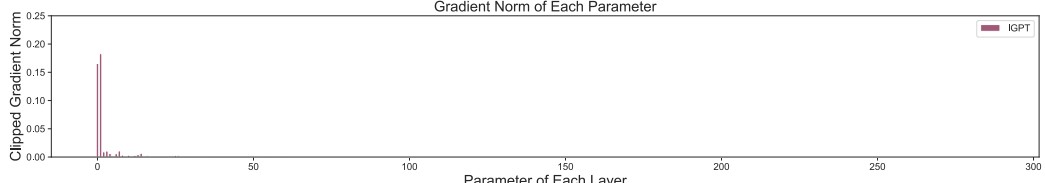

**Figure 66:** Gradient norm of iGPT's each parameter at epoch 1. (HalfCheetah, Seed = 42)

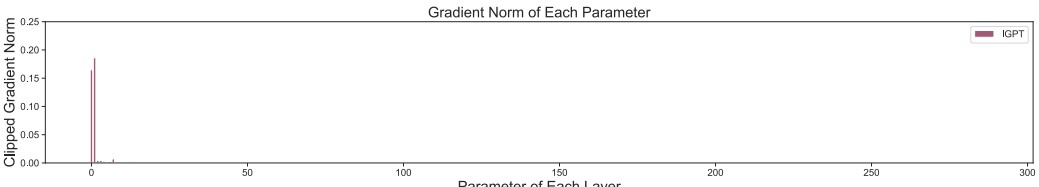

**Figure 67:** Gradient norm of iGPT's each parameter at epoch 1. (Walker2D, Seed = 42)

## J.5 Fine-Tuning with No Context Information

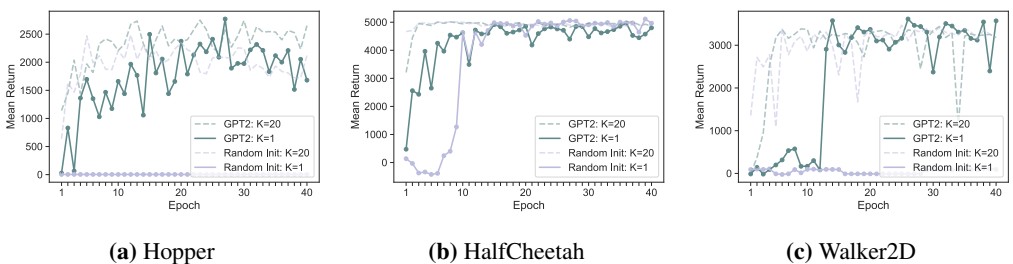

(a) Hopper      (b) HalfCheetah      (c) Walker2D

**Figure 68:** Mean return throughout fine-tuning when access to the context information is prohibited (seed = 42).

**Table 4:** Normalized return of $K = 1$ (seed = 666 & 42).

| Dataset | Environment | GPT2 | Random Init |
|---------|-------------|------|-------------|
| | Hopper | $81.3 \pm 2.5$ | $-0.3 \pm 0.1$ |
| Medium | HalfCheetah | $47.9 \pm 0.1$ | $49.2 \pm 0.1$ |
| | Walker 2D | $71.2 \pm 2.2$ | $35.6 \pm 33.5$ |

## J.6 More In-Depth Analysis of Context Dependence

### J.6.1 Replacement by the Pre-Trained Block

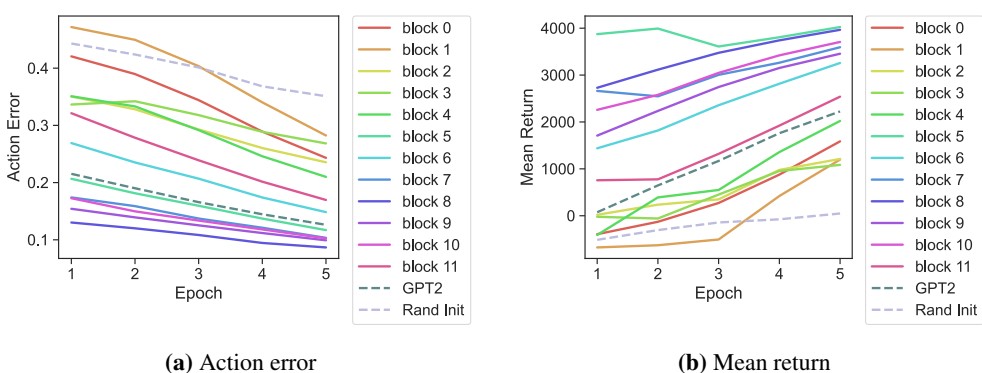

**(a)** Action error        **(b)** Mean return

**Figure 69:** Learning curve when only a block is pre-trained (HalfCheetah, Seed = 666).

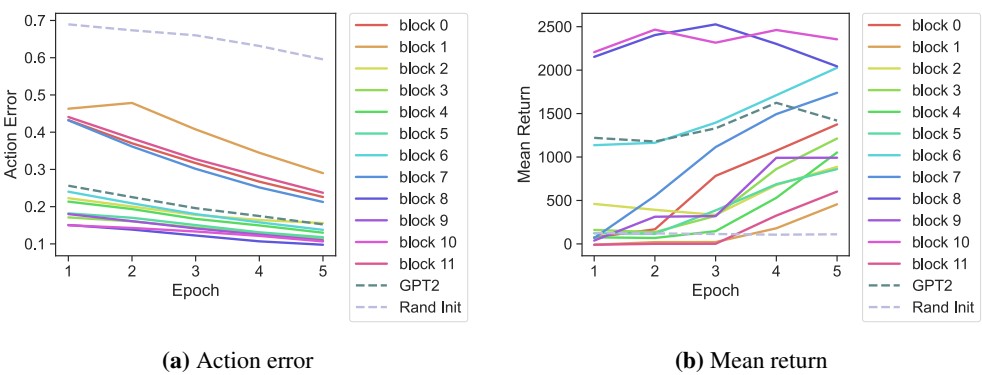

**(a)** Action error        **(b)** Mean return

**Figure 70:** Learning curve when only a block is pre-trained (Walker2D, Seed = 666).

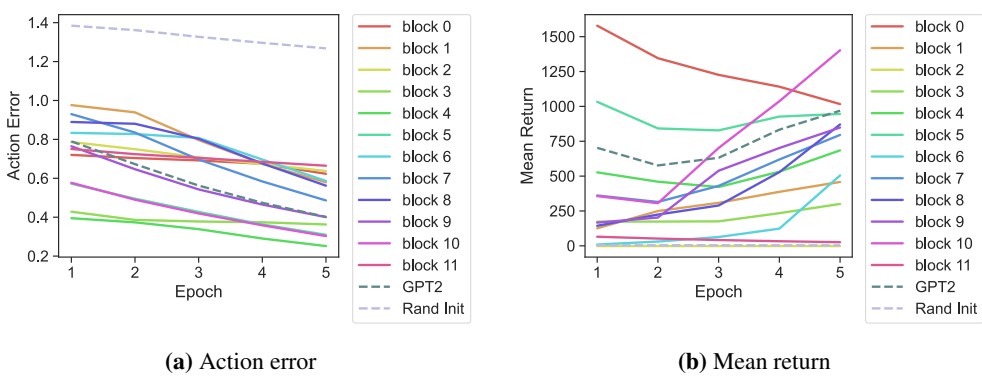

**(a)** Action error        **(b)** Mean return

**Figure 71:** Learning curve when only a block is pre-trained (Hopper, Seed = 42).

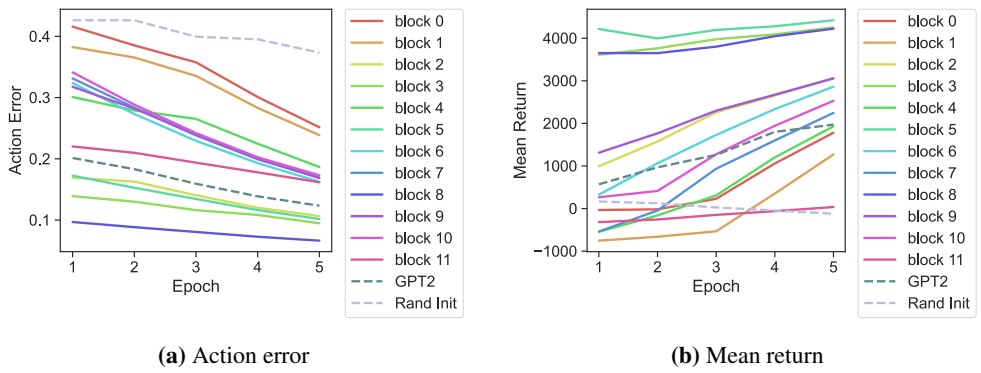

(a) Action error

(b) Mean return

**Figure 72:** Learning curve when only a block is pre-trained (HalfCheetah, Seed = 42).

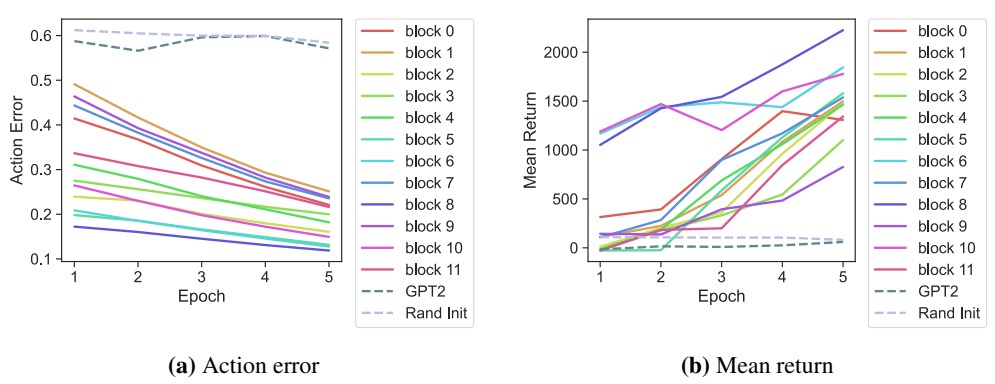

(a) Action error

(b) Mean return

**Figure 73:** Learning curve when only a block is pre-trained (Walker2D, Seed = 42).

### J.6.2 Attention Distance Analysis

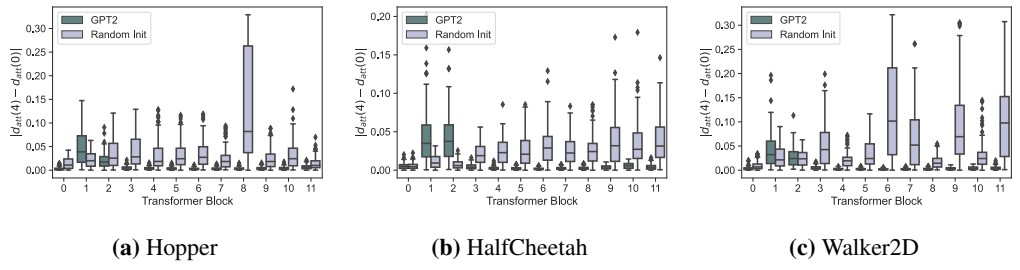

(a) Hopper

(b) HalfCheetah

(c) Walker2D

**Figure 74:** Attention distance gap between epoch 0 and epoch 1.

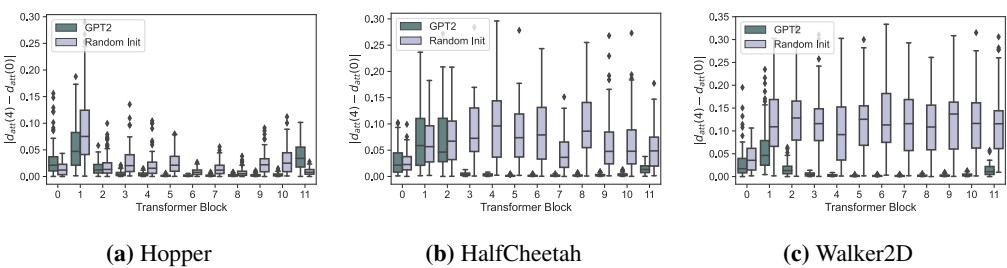

(a) Hopper

(b) HalfCheetah

(c) Walker2D

**Figure 75:** Attention distance gap between epoch 0 and epoch 10.

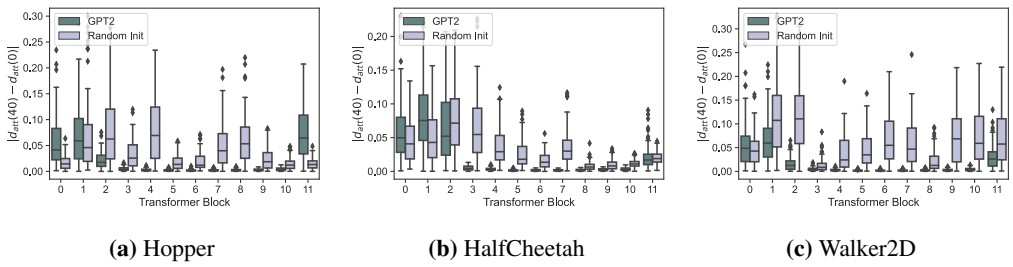

**(a)** Hopper        **(b)** HalfCheetah        **(c)** Walker2D

**Figure 76:** Attention distance gap between epoch 0 and epoch 40.

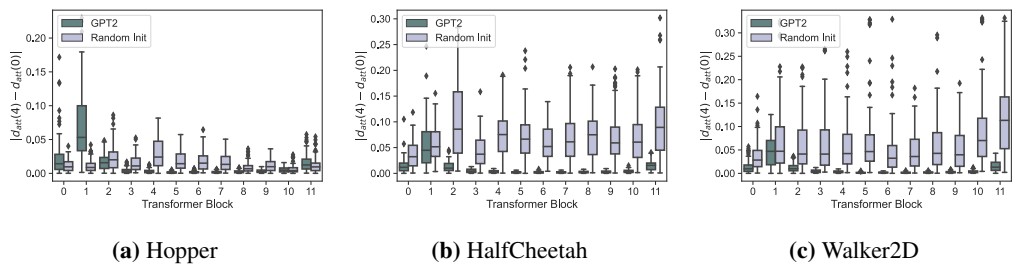

**(a)** Hopper        **(b)** HalfCheetah        **(c)** Walker2D

**Figure 77:** Attention distance gap between epoch 0 and epoch 4 (seed = 42).

# K   Details for Computation

The computational environment for our experiments is as follows:

- GPU: NVIDIA GeForce GTX TITAN X $\times$ 2GPU.
- CPU: Intel Xeon E5-2620 v3 2.4 GHz, 6 Cores $\times$ 2CPU.
- software: Ubuntu 20.04.4, GCC 9.4.0, CUDA 11.6.

The computational time of fine-tuning for 40 epochs is around from 24 to 72 hours for context length $K = 20$ and around from 7 to 48 hours for context length $K = 1$ per run.

# L   Licences of Assets Used for Our Experiments

## L.1   Dataset

We use the offline RL dataset of Mujoco in D4RL [38]. The license of the dataset in D4RL is a Creative Commons Attribution 4.0 License (CC BY). Thus, we can use this dataset with no consent as long as we follow the term of use.

## L.2   Code

The code we used does not require special consent from authors as long as we follow the terms of use. Their licenses are as follows:

- https://github.com/machelreid/can-wikipedia-help-offline-rl: MIT Licence.
- https://github.com/rail-berkeley/d4rl: Apache License 2.0.
- https://github.com/google-research/google-research/tree/master/representation_similarity: Apache License 2.0.
- https://github.com/gtegner/mine-pytorch: MIT License.

## M   Note on the Role of Each Sub-Sections in Section 5

Sections 5.1 and 5.2 show what may *not* be a cause of the good performance. In other words, these sections eliminate some seemingly possible causes of good performance of language-pre-trained models, respectively. These analyses further highlight the importance of attention distance for performance. Without the analysis of Section 5.1, we cannot exclude the possibility that good performance comes from re-using pre-trained representation *as well*, which is common in the uni-modal case as we explained in Section 5.1. Similarly, without the analysis of Section 5.2, the performance can benefit from fitting better into training data *as well*. In that case, it would be less clear whether utilizing some prior knowledge is solely important. Sections 5.4, 5.5, 5.6.1, and 5.6.2 discuss what may be a cause of the good performance, as summarized at the end of Section 5. Section 5.3 provides complementary findings for these sections. For example, the existence of some unchanged parameters implies the possibility that some information is re-used, while the relatively larger change in only shallower layers of pre-trained models partially supports that these unchanged parameters do not seem to contradict with changed representation observed in Section 5.1, as explained in Appendix N.

## N   Note on Relationship Between Parameter Change and Representation Change

In Section 5.1 we say that *representation* in some layers of the pre-trained model changes, while in Section 5.3 we say that *parameters* of them in some layers do not change that much, or vice versa for the randomly initialized model. These observations of Sections 5.1 and 5.3 are not necessarily contradictory because the change of parameters in layer $\ell$ does not correspond one-to-one with that in representation in the layer.

First, the representation of layer $\ell$ is affected by all parameters and input data prior to layer $\ell$. So, even if the parameters of layer $\ell$ have not changed, if the parameters before layer $\ell$ have changed, the representation of layer $\ell$ may change. Second, since not all of the parameters of a neural network necessarily contribute to the output, the output of a layer may not change even if some parameters of that layer change. For example, if the input value to ReLU is negative, the output will remain 0 no matter how much the parameters involved change. So, for example, it is possible that even if the parameters of the $\ell$-layer change, the representation of the $\ell$-layer remains the same. Another example is when the values of two weights $w_1^\ell$ and $w_2^\ell$ in layer $\ell$ do not change the output of the vanilla feed-forward network because of its symmetry.

Another possible cause of this observation unique to the current analysis is that we use CKA to measure representational similarity. CKA is designed to be invariant to some transformations on the representation matrix [39]. Thus, different representation matrix is regarded as the same under these transformations even when parameters are changed.

## O   Note on Why Better Action Prediction Does Not Always Result in Better Return

In Section 5.2, we explained that randomly initialized models seem to predict action better, while in Table 1, they perform worse than GPT2. We will add a possible explanation of why better action prediction not necessarily comes to a better return.

The most likely cause of this observation is the current problem setup. As mentioned in Appendix B, the *medium* dataset is early stopped and thus does not necessarily converge to an optimal policy. This means that if the model accurately learns trajectory and predicts the next action, it does not always mean that it is the best action. Hence, we can see that a low action error does not necessarily mean a large mean return. However, We do not believe that the use of medium will hurt the validity of the analysis using this data since this dataset is not that pathological.

Another cause might be related to mutual information. The current analysis of mutual information shows that the representation of the hidden layer of the randomly initialized model holds more information as well of the input as well as the labels. Since the neural net representation is a vector representation if a single vector contains both types of information, these types of information

might have been mixed. This may make it difficult to properly utilize only the label information, making it harder to accurately predict the action. Another possibility is that the pre-trained model and the randomly initialized model differ in terms of which input token type (return-to-go, state, or action) information they encode. For example, in Fig. 2 (b), the language pre-trained model encodes information uniformly from the same token type: the hidden representation with high mutual information is the representation corresponding to the input from the triangles, i.e., the part corresponding to the return-to-go. On the other hand, in the randomly initialized models, there are variations (triangle, circle, and cross). It was noted in a previous study that attention weight was strong between the same token types [9]. Therefore, perhaps the pre-trained model might encode less total information, but the amount of information *effectively* utilized is not that different. Finally, as pointed out in the explanation about the limitation, the limitations of mutual information as a metric might be a cause.