# OpenReview forum: "On the Effect of Pre-training for Transformer in Different Modality on Offline Reinforcement Learning"
_NeurIPS.cc/2022/Conference — NeurIPS 2022 Accept_

### Official Review · Reviewer_JfVS · 2022-07-11

**Rating:** 7
**Confidence:** 4
**Soundness:** 3 good
**Presentation:** 2 fair
**Contribution:** 3 good

**Summary:**

This paper presents an analysis of how the pretraining-finetuning paradigm works for Transformer models in the context of reinforcement learning. Pretrained Transformer models trained on two different modalities (GPT-2 for language, and iGPT for image) are applied to Mujoco offline RL tasks - Hopper, HalfCheetah and Walker2D. The paper examines how the representations change during finetuning - which is achieved by using several techniques such as activation similarity, distance between pretrained and finetuned weights, mutual information between training data and learned representations, gradient norms and magnitudes etc. Through this analysis, the paper uncovers some findings: primarily, how different language and image-based pretrainings fare during finetuning for RL, how different layers of the transformers exhibit different magnitudes of changes - demonstrating how pretraining knowledge trickles down to finetuning, and potentially why iGPT underperforms compared to the language GPT. It is also shown that the language GPT version achieves reasonable performance on RL tasks even under the case of constrained context length, thus highlighting the fact that it might already be aware of context-related information from pretraining.

**Questions:**

- Have the authors experimented with changing the gradient norm clipping value for iGPT? It seems unlikely that it would cause a major shift in the performance of iGPT, but it might help with understanding the role of gradient magnitudes and gradient confusion in the final performance of these models.

- Why does the random init model with no context (K = 1) perform significantly worse than the pretrained ones only in Hopper? For the other two environments, it actually performs a bit better than the pretrained ones.

**Limitations:**

Mostly yes. The analysis could change a bit if the models were to be trained on a different corpus, such as not language or images, but rather trajectories from different RL tasks in a multi-task learning fashion.

**Strengths And Weaknesses:**

Strengths:

- The paper presents a very interesting and thought-provoking analysis of representation dynamics between pretraining and finetuning. Given the current interest in large foundational models being the starting point for perception and perception-action, the paper is a good contribution to the field, and can aid in understanding how large Transformers can transfer between different domains.
- The paper identifies a set of relevant techniques to perform analysis of representation similarity, training dynamics through gradient analysis etc. which can be useful to apply to other large models as well. By applying these techniques to large Transformer models, the paper extends the findings in papers such as [1] to understand why or why not these models perform well at RL tasks.

Weaknesses:

- I believe the paper can benefit from a bit of reframing and rewrite. After reading through the paper, some of the initial points end up sounding a bit confusing. For instance, the paper states in several places that "pretrained models largely change their representation", and then "pretrained models do not change parameters much". It is a bit unclear exactly what is leading to what in terms of performance.

1. In terms of parameters, for the pretrained models, it seems that the only significant changes are in the shallow layers, and to a lesser extent, in the layer norms (section 5.1, section 5.3). Whereas for the randomly initialized model, most of the changes are in the layer norms. Does this still explain the consistency of representation in the shallow to middle layers of random init? (section 5.1)

2. The analysis in 5.2, which investigates the amount of label information being encoded can benefit from a little more clarity. As it stands, it looks like the randomly initialized model does a better job at predicting the appropriate action given the states. Yet, random init does noticeably worse than GPT-2 on Hopper medium in table 2 or figure 9(a) (with K=20). It would be good to shed some light on exactly which objective ends up mattering the most for performance.

3. In section 5.4, the paper hypothesizes that the fact that the gradient confusion is higher for the pretrained models, that makes them hard to train which explains the smaller changes in parameters. But this does not again seem to affect the performance in a detrimental way.

- While I acknowledge that the analysis is meant build upon the findings from [1], one can't help but wonder that because reinforcement learning performance requires encoding state transitions, whether models that are trained on large corpuses of RL data (similar to the concepts in multi-game decision transformer, or Gato) would fare better at RL and shed light on pretraining-finetuning that's directly

- The analysis being done in the paper is quite extensive and nuanced. It would also be useful to have a summary of the findings at the end of the results section that summarizes the main takeaway points from all the subsections combined, otherwise it can be hard to keep track. Similarly, the paper would benefit from a summary of the baseline results in the main text itself (moving Table 2 from appendix to section 5).

[1] Machel Reid, Yutaro Yamada, Shixiang Shane Gu, "Can Wikipedia help Offline Reinforcement Learning?"

---

> ### Author Response · Authors · 2022-08-01
> **Response to Reviewer JfVS (1/5)**
>
> Thank you very much for reading our manuscript so carefully and giving us so much constructive feedback. We would like to respond to each of your points below.
>
> > “the paper can benefit from a bit of reframing and rewrite”
>
> > “It is a bit unclear exactly what is leading to what in terms of performance”
>
> > “It would also be useful to have a summary of the findings at the end of the results section that summarizes the main takeaway points from all the subsections combined”
>
> > “exactly which objective ends up mattering the most for performance”
>
> We appreciate your significant input to improve the readability of our paper. As you suggested, we have added a summary of findings at the end of the results section. We have paid particular attention to the implications of each finding for the performance findings. The added summary is below:
>
> - 5.1：Re-using representation is not the cause of the good performance of GPT2.
> - 5.2：Fitting to data better is not the cause of the good performance of GPT2.
> - 5.3：The good performance of GPT2 might come from some unchanged parameters.
> - 5.4：Gradient clipping and large gradient might be a cause of the bad performance of iGPT.
> - 5.5 ：GPT2 can learn efficiently even without the context provided.
> - 5.6.1：Even a single pre-trained Transformer block makes training more efficient.
> - 5.6.2：A cause of the good performance of GPT2 is a pre-acquired way to use context.
>
> Additionally, we have added explanations on the relationships between findings of each section in Appendix N to make it easier to get how each section contributes to our claim. We believe that these re-framing and re-writing would help to understand which finding lead to what in that of performance.
>
> > “the paper would benefit from a summary of the baseline results in the main text itself (moving Table 2 from appendix to section 5).”
>
> We have moved the table of baseline results (Table 2 in Appendix 1) to the main text (Table 1 in Section 4). Thank you for your suggestions to make our manuscript clearer.

---

> > ### Author Response · Authors · 2022-08-01
> > **Response to Reviewer JfVS (2/5)**
> >
> > > “In terms of parameters, for the pretrained models, it seems that the only significant changes are in the shallow layers, and to a lesser extent, in the layer norms (section 5.1, section 5.3). Whereas for the randomly initialized model, most of the changes are in the layer norms. Does this still explain the consistency of representation in the shallow to middle layers of random init? (section 5.1)”
> >
> > Thank you very much for pointing out the part that is difficult to understand. We may not have accurately captured the intent of your question, if so we would appreciate it if you could point out if we have missed the point of my response.
> >
> > First, we would like to respond to your point about the random initialization model. First of all, let us reiterate that the higher the value of CKA, the higher the degree of similarity. Thus, what the figure in 5.1 shows is that the expression of the shallow to middle layers, especially that of layer norm has “not” changed. This may have been a point of misunderstanding because our notation was not clear. Thus we have made it clear in Section 5.1. Since the shallow to middle layers have not changed, we have described in the paper that the representations of randomly initialized model have a higher similarity to that of the initial state from the middle to the shallow layers.
> >
> > Next, we would like to respond to your point about pre-trained models. For the pre-trained model, the CKA of the pre-trained model in Figure 1 in Section 5.1 also has a few high values in the shallow layers. This again indicates that the changes of representations are relatively "small" in this part of the model, as we have just explained above. As you said in the review, the figures in Section 5.3 shows that most of the changes in the parameters of the pre-trained model occur in the shallow layer, in the sense of the l2 distance of the parameters.
> >
> > This observation of 5.1 and 5.3, i.e., that the parameters change and the representation does not change or vice versa, is not necessarily contradictory. First, the representation of layer $\ell$ is affected by all parameters and input data before layer $\ell$. Hence, even if the parameters of layer $\ell$ have not changed, if the parameters before layer $\ell$ have changed, the representation of layer $\ell$ may change. Second, since not all of the parameters of a neural network necessarily contribute to the output, the output may not change even if some parameters change. For example, if the input value to ReLU is negative, the output will remain 0 no matter how much the parameters involved change. Therefore, for example, it is possible that even if the parameters of the $\ell$-layer change, the representation of the $\ell$-layer remains the same. We have added this complementary explanation of this relationship between parameters and representation in Appendix O. We are not sure if this is the answer to your question, but we would appreciate it if you could point out if we misunderstood your intention.

---

> > > ### Author Response · Authors · 2022-08-01
> > > **Response to Reviewer JfVS (3/5)**
> > >
> > > > “it looks like the randomly initialized model does a better job at predicting the appropriate action given the states. Yet, random init does noticeably worse than GPT-2 on Hopper medium”
> > >
> > > Thank you for your valuable feedback. Of course, in general, decreasing action error is associated with increasing mean return (e.g., as in Figure 9 a, where a decrease in action error and an increase in mean return seem to be generally related as a trend). However, in the current problem setup, it appears that more accurate action prediction does not always lead to higher performance improvement. For example, looking at Figure 9, it seems that block 9 (the pink line in Figure 9 a) with the lowest action error does not achieve the highest mean return (the pink line in Figure 9 b).
> > >
> > > The most likely cause of this observation is the current problem setup. The D4RL mujoco task used as offline RL data for our analysis has *expert*, *medium*, *medium-replay*, *medium-expert*, and *random* datasets. The *expert* dataset is the one trained by Soft Actor-Critic, *medium* is the one partially trained by Soft Actor-Critic and was stopped early, *medium-replay* is the one accumulated in the replay buffer before the model reached medium’s level, and *medium-expert* is a mixture of medium and expert results. The *random* dataset is the trajectory collected by the random policy. We conducted our experiments on the *medium* data set since it was used by the previous study [1] and our analysis is based on the observations of this previous study. As mentioned earlier, this dataset is early stopped and so does not necessarily converge to an optimal policy. This means that if the model accurately learns trajectory and predicts the next action, it does not always mean that it is the best action. Thus, we can see that a low action error does not necessarily mean a large mean return. We do not believe that the use of medium will hurt the validity of the analysis using this data since this dataset is not that pathological as explained above. We have added the explanation of the data in more detail in Appendix B.
> > >
> > > Another cause might be related to mutual information. The current analysis of mutual information shows that the representation of the hidden layer of the randomly initialized model holds more information as well of the *input* as well as the labels. Since the neural net representation is a vector representation if a single vector contains both types of information, these types of information might have been mixed. This may make it difficult to properly utilize only the label information, making it harder to accurately predict the action. Another possibility is that the pre-trained model and the randomly initialized model differ in terms of which input token type (return-to-go, state, or action) information they encode. For example, in Fig. 2 (b), the language pre-trained model encodes information uniformly from the same token type: the hidden representation with high mutual information is the representation corresponding to the input from the triangles, i.e., the part corresponding to the return-to-go. On the other hand, in the random init models, there are variations (*triangle*, *circle*, and *cross*). It was noted in a previous study that attention weight was strong between the same token types [1]. So, perhaps the pre-trained model might encode less total information, but the amount of information effectively utilized is not that different. Finally, as pointed out in the explanation about the limitation, the limitations of mutual information as a metric might be a cause. We have included these discussion above in the Appendix P.
> > >
> > > In any case, what we can say from the analysis in Section 5.2, at least, is that the reason pre-trained models work well does not seem to be because the models acquire more information about the data. In Section 5.6, we found that preserving the attention distance enables efficient learning. However, this alone does not rule out the possibility that acquiring more data is "also" beneficial. We believe that the results in Section 5.2 lowers that possibility and highlight the possibility that model performance is uniquely dependent on attention distance.
> > >
> > > Thank you for providing your thought-provoking question. It has given us an opportunity to delve into the results of the analysis in Section 5.2.

---

> > > > ### Author Response · Authors · 2022-08-01
> > > > **Response to Reviewer JfVS (4/5)**
> > > >
> > > > > “models that are trained on large corpuses of RL data (similar to the concepts in multi-game decision transformer, or Gato) would fare better at RL and shed light on pretraining-finetuning”
> > > >
> > > > > “The analysis could change a bit if the models were to be trained on a different corpus, such as not language or images, but rather trajectories from different RL tasks in a multi-task learning fashion”
> > > >
> > > > As you say, it is expected to perform better with pre-training in RL and it is a very important direction to go into. We would like to do this in the future. We have noted in the discussion section that this is important for future work.
> > > >
> > > > One of the contributions of our work is that we have utilized analysis methods that can be applied to these RL pre-trained models as well, as you pointed out. For example, by examining CKA, we can determine whether and in which layer feature-reuse occurs for RL pre-trained model. One of the significances of analyzing multimodality is to deepen our understanding of the limitations and applicability of the foundation model, as you pointed out. Another benefit we believe is to clarify information that can be used across different modalities, in order to consider what kind of inductive bias we should install to the general-purpose models.
> > > >
> > > > > “Have the authors experimented with changing the gradient norm clipping value for iGPT? It seems unlikely that it would cause a major shift in the performance of iGPT, but it might help with understanding the role of gradient magnitudes and gradient confusion in the final performance of these models.”
> > > >
> > > > We did not do at submission any experiments to change gradient clipping. Because we thought your point was essential, we took it and performed a simple experiment.
> > > >
> > > > The experiments of the previous study [1] and our experiments both used a Pytorch function called `torch.nn.utils.clip_grad_norm_` for gradient clipping. This function divides each gradient by the “total” norm of all gradients and multiplies the clipping constant. However, as we pointed out in the paper, the gradient norm values are dominated by only a few parameters (the layer normalization layer in the first block). This large norm affects the normalization of all gradients, decreasing the informational value of most gradients. Therefore, we hypothesized that one of the possible reasons why iGPT is difficult to learn can be the use of the `torch.nn.utils.clip_grad_norm_` function. Thus we trained the image-pre-trained model without using gradient clipping. Since we didn't have time, we only trained for 10 epochs (instead of the original 40 epochs).
> > > >
> > > > The results showed that, as you said, eliminating gradient clipping did not immediately solve the catastrophic performance at least up to 10 epochs. At the same time, we did find that the learning process seemed to be more stable and efficient than before the clipping was applied. We also found that the performance seemed to improve, albeit very slightly. Based on these results, we have weakened the statement from "too large gradients may be the cause of the catastrophic performance" to "large gradient and gradient clipping may be one of the causes of the poor performance". We then have clearly stated that the essential cause of the catastrophic performance should be pursued, and added the results and procedure of the simple experiment explained above in Appendix G.3. Thank you very much for your very significant suggestions.
> > > >
> > > > For gradient confusion, a previous study claims that widening layers lessens the impact of gradient confusion [2]. Thus, checking if employing a wider network when pre-training mitigates the difficulty of the training might help to understand the importance of gradient confusion in performance. Since we do not have the time to re-train the wider network we leave this as future work. Instead, we have weakened the statement of the paper to be consistent with our findings.

---

> > > > > ### Author Response · Authors · 2022-08-01
> > > > > **Response to Reviewer JfVS (5/5)**
> > > > >
> > > > > > “Why does the random init model with no context (K = 1) perform significantly worse than the pretrained ones only in Hopper?”
> > > > >
> > > > > Thank you for your question. We speculate that this may be because *how much of the range of context that needs to be looked at* changes depending on the data set. For example, prior studies on decision transformer reported that the effect of context varied depending on the task [3]. Improvement by context means that the context was important information for solving the task. Thus, we believe that Hopper may be the dataset that requires a longer look at context than the other two data sets.
> > > > >
> > > > > As a test of this hypothesis, we randomly sampled a batch sample and calculated the mutual information between action and state or return-to-go at the same time step and compare them between different environments; note that this is the mutual information between the data, not between the data and representation. The higher the value of the mutual information, the higher the mutual dependence between state or return-to-go and action at the same time step. Hence, higher mutual information suggests that the model could predict action better only from the information at the current time step.
> > > > >
> > > > > As a result of this analysis, we found that mutual information between return-to-go and action was smaller for Hopper than for the others, though that between action and state does not differ between different environments that much. Prior research has indicated that return-to-go information seems to be important for prediction [1]. Hence, we can say that models have to use more information from the other steps to solve the Hopper task than models do for other environments. This result supports our hypothesis above. We will summarize the procedure and the result in the Appendix H.2.
> > > > >
> > > > > References
> > > > >
> > > > > [1] Machel Reid, Yutaro Yamada, and Shixiang Shane Gu. Can wikipedia help offline reinforcement learning? arXiv preprint arXiv:2201.12122, 2022.
> > > > >
> > > > > [2] Karthik Abinav Sankararaman, Soham De, Zheng Xu, W Ronny Huang, and Tom Goldstein. The impact of neural network overparameterization on gradient confusion and stochastic gradient descent. In International conference on machine learning, pages 8469–8479. PMLR, 2020.
> > > > >
> > > > > [3] Lili Chen, Kevin Lu, Aravind Rajeswaran, Kimin Lee, Aditya Grover, Michael Laskin, Pieter Abbeel, Aravind Srinivas, and Igor Mordatch. Decision transformer: Reinforcement learning via sequence modeling. In A. Beygelzimer, Y. Dauphin, P. Liang, and J. Wortman Vaughan, editors, Advances in Neural Information Processing Systems, 2021.

---

> > ### Comment · Reviewer_JfVS · 2022-08-08
> > **Response to author rebuttal**
> >
> > I would like to thank the authors for their extensive rebuttal response that tackled many of my questions. Primarily, I appreciate their modifications to the paper that better highlight the findings from their analysis; and the insightful responses regarding the performance of the models specifically relating to discrepancy between label prediction and actual returns of a policy; the effect of context etc. which are added to the appendices. Given the burgeoning popularity of Transformers for control/RL tasks, I think the extent of the analysis performed in the paper as well as the discussion together form a good contribution to the community. Furthermore, the collection of techniques used in this work can be applicable for other domains where the pretraining-finetuning paradigm is common. I am happy to upgrade my score, and would recommend the authors to open source their code so others can build upon this work.

---

> > > ### Author Response · Authors · 2022-08-09
> > > **Thank you for your response!**
> > >
> > > We deeply appreciate the positive feedback on our response. We would gladly open source the code for the research community.

---

### Official Review · Reviewer_GdNs · 2022-07-12

**Rating:** 7
**Confidence:** 4
**Soundness:** 3 good
**Presentation:** 3 good
**Contribution:** 3 good

**Summary:**

In this work, the authors empirically investigate how pre-training on language and image data can affects fine-tuning of Transformers on offline reinforcement learning tasks. Previous studies reported that, for offline RL tasks, pre-training a Transformer-based model on image data either maintains or helps performance, whereas pre-training on vision data deteriorates performance. To understand the effect of fine-tuning, the paper compares between models that are randomly initialized, pre-trained with language data (GPT2), and with image data (iGPT). For that purpose they used GPT2 architecture. For offline RL tasks, the paper employs medium datasets of HalfCheetah, Walker2d, and Hopper environments. The authors analyze various network artifacts, like how layer activation and model weights change during training, initials gradient norms, also what type of language-trained model learns and utilizes for down-stream tasks, and others. The conclusion roughly is that, pre-trained model parameters do not change much during fine-tuning, though their layer representations change largely. Pre-trained models also do not acquire more knowledge about the input or the label during fine-tuning. The authors also hypothesize with empirical proofs that, the image-pre-trained model fails probably because of large gradients, and also that language pre-training probably gives the Transformer context-like information that can be taken advantage of during down-stream tasks.

**Questions:**

- Lines 181-182, "does not 182 encode more information about the input or the label". This may be because of limitation of mutual information, as the authors themselves pointed out in line 326. So should take the finding that, pre-trained Transformers do not acquire more knowledge about the input or the label during fine-tuning, with pinch of salt.
- In section 5.3, we can see parameters between pre and post-fine-tuning have changed for random initialization. Then in section 5.1, why representations for random initialization did not change much, i.e., the CKA scores for representations from random initialization are high?
- In line 263, I am guessing, fine-tune without context.
- In line 281, the authors noted that "other than a particular block". But by considering results from all three datasets, it is more like all blocks can contribute only in varying amount. That is, the observation is more in line with lines 303-304.

**Limitations:**

The limitations about the validity of the metrics used, is already pointed out in line 326.

**Strengths And Weaknesses:**

Strengths:
- The topic investigated in this work is important and has room to be explored.
- The paper presents a well-written, detailed study.
- Limitations, mainly from the validity of the metrics used, is clearly pointed out.

---

> ### Author Response · Authors · 2022-08-01
> **Response to Reviewer GdNs**
>
> Thank you for your positive feedback on our paper. We would like to answer the comments you have raised below.
>
> > “does not 182 encode more information about the input or the label". This may be because of limitation of mutual information”
>
> As you mentioned in the review, we agree that this conclusion can come from the limitations of mutual information. We again emphasized in the text that this conclusion can be affected by those limitations. Thank you for pointing this out.
>
> > “In section 5.3, we can see parameters between pre and post-fine-tuning have changed for random initialization. Then in section 5.1, why representations for random initialization did not change much”
>
> We appreciate you asking the question on this point. We suspect that this is because neural networks in general do not necessarily change its representation when the parameters are changed. For example, if the input value to ReLU is negative, the output will remain 0 no matter how much the parameters involved change. Hence, for instance, it is possible that even if the parameters of the $\ell$-layer change, the representation of the $\ell$-layer remains the same. Another example is that when the values of two weights $w_1^\ell$ and $w_2^\ell$ in layer $\ell$ do not change the output of the vanilla feedforward network because of its symmetry.
>
> Another possible cause of this observation unique to the current analysis is that we use CKA to measure representational similarity. CKA is designed to be invariant to some transformations on the representation matrix [1]. Thus, different representation matrix is regarded as the same under these transformations even when parameters are changed. We have added this complementary explanation of the relationship between parameters and representation in Appendix O.
>
> > “In line 263, I am guessing, fine-tune without context.”
>
> You are correct. We added the explicit statement in the paper that it’s fine-tuning without context. Thank you for the comment to improve clarification.
>
> > “it is more like all blocks can contribute only in varying amount. That is, the observation is more in line with lines 303-304.”
>
> Your suggestion sounds more appropriate than our current description. We have rewritten the statement to “we find that the information improving learning efficiency is preserved in all blocks only in the varying amount and that is probably context-related” so that we reflect your comment. Thank you so much for your recommendations for better wording.
>
> References
>
> [1] Simon Kornblith, Mohammad Norouzi, Honglak Lee, and Geoffrey Hinton. Similarity of neural network representations revisited. In International Conference on Machine Learning, pages 3519–3529. PMLR, 2019.

---

### Official Review · Reviewer_mvwR · 2022-07-12

**Rating:** 5
**Confidence:** 4
**Soundness:** 2 fair
**Presentation:** 2 fair
**Contribution:** 2 fair

**Summary:**

The paper presents an empirical study of language pre-training (GPT-2), image pre-training (iGPT), and training from scratch for offline RL. The study looks at the differences in activations, parameters, and gradients as well as the impact of the context length.

**Questions:**

Suggestions:
- It would be good to confirm that the observed trends hold when using more than one seed


**Limitations:**

The paper discusses limitations and potential negative societal impact.

**Strengths And Weaknesses:**

Strengths:
- The study considers an interesting question
- The experiments show interesting results (e.g., impact of context)

Weaknesses:
- The paper considers the number of aspects (e.g., activations, parameters, etc.) and reports interesting observations. However, it feels that it often stops too early and presents a hypothesis without trying to explore it further. For example, in the case of the gradient analysis the paper suggests that iGPT does not train well due to large gradients in early layers. It would be nice to explore this further and see if, e.g., this observation can inform a better training recipe for offline RL with iGPT pre-training.
- Writing and presentation are sometimes a bit hard to follow

---

> ### Author Response · Authors · 2022-08-01
> **Response to Reviewer mvwR (1/2)**
>
> Thank you for giving up your time to review our manuscript. Below we would like to respond to your individual feedback one by one.
>
> > “it feels that it often stops too early and presents a hypothesis without trying to explore it further”
>
> Thank you for your valuable feedback. We have added analysis to those where the further exploration could bring more insights. For some of those that were not conducted an additional experiment, we have added explanations that may lead to practical insights. In particular, we have added new analysis of the effect of gradient clipping on performance (Appendix G.3) and why randomly initialized model struggle to learn Hopper with no context (Appendix H.2). We have also added notes in Appendix O and P regarding other results where further exploration of the observation would benefit or where the results cross over into the results of each section to provide implications (e.g., the relationship between changing parameters and changing expressions).
>
> > “in the case of the gradient analysis the paper suggests that iGPT does not train well due to large gradients in early layers. It would be nice to explore this further and see if, e.g., this observation can inform a better training recipe for offline RL with iGPT pre-training.”
>
> We thought your feedback is very significant. Thus, we have conducted an experiment to provide some implications for better training.
>
> The experiments of the previous study [1] and our experiments both used a Pytorch function called `torch.nn.utils.clip_grad_norm_` for gradient clipping. This function divides each gradient by the total norm of all gradients and multiplies the clipping constant. However, as we pointed out in the paper, the gradient norm values are dominated by only a few parameters (the layer normalization layer in the first block). This large norm affects the normalization of all gradients, decreasing the informational value of most gradients. Therefore, we hypothesized that one of the possible reasons why iGPT is difficult to learn can be the use of the `torch.nn.utils.clip_grad_norm_` function. Thus we trained the image-pre-trained model without using gradient clipping. Since we didn't have time, we only trained for 10 epochs (instead of the original 40 epochs).
>
> The results showed that, although eliminating gradient clipping did not immediately solve the catastrophic performance, the learning process seemed to be more stable and efficient than before the clipping was applied. We also found that the performance seemed to improve, albeit very slightly. Thus, we might say that removing gradient clipping might be a recipe for better training. We have added the details in Appendix G.3. Thank you very much for your constructive feedback.
>
> Although we did not confirm its validity experimentally, another possible way to improve training efficiency would be smoothing the loss landscape because a loss landscape with large gradients dominated by a few parameters is thought to have a distorted topography. Given that batch normalization is known to smooth loss landscapes [2], using PowerNorm [3], which allows batch normalization to be applied to the Transformer, may make the learning process more stable.
>
> > “Writing and presentation are sometimes a bit hard to follow”
>
> Thank you for your comment on this important point. We have taken your points as crucial and have made changes in the writing style of the manuscript and added supplementary explanations. We suspected that part of the difficulty in understanding the text might be due to the difficulty in understanding the main argument of each section. Therefore, we summarize the main takeaways of Section 5 in bullet point format at the end of the section. We also thought that the difficulty in understanding how the claims in these sections relate to each other may have added to the difficulty of reading them. Thus, we have added a supplemental description in Appendix N that briefly explains the role of each section (Sections 1 - 5) of this paper. In addition to that, we have added explanations for sentences where the meaning is not clear.

---

> > ### Author Response · Authors · 2022-08-01
> > **Response to Reviewer mvwR (2/2)**
> >
> > > “It would be good to confirm that the observed trends hold when using more than one seed”
> >
> > We appreciate you for pointing out this very important point. To reflect your feedback, we have done some experiments with the additional seed (seed = 42 in the manuscript) as well and confirmed that our conclusions are largely unaffected by these results. Specifically, we have added in Appendix J results from the new seed for the following analyses:
> >
> > - CKA between pre and post-fine-tuning (Figs. 25 - 27)
> > - Mutual information between hidden representation and data (Fig. 46)
> > - Parameter similarity analysis (Figs. 51 - 56)
> > - Gradient analysis (Figs. 59 - 67)
> > - Replacement by the pre-trained block (Figs. 71 - 73)
> > - Attention distance analysis (Fig. 77)
> >
> > We believe that we have made our manuscript more convincing thanks to your suggestion. We appreciate again you for your valuable feedback.
> >
> > References
> >
> > [1] Machel Reid, Yutaro Yamada, and Shixiang Shane Gu. Can wikipedia help offline reinforcement learning? arXiv preprint arXiv:2201.12122, 2022.
> >
> > [2] Shibani Santurkar, Dimitris Tsipras, Andrew Ilyas, and Aleksander Madry. How Does Batch Normalization Help Optimization?. In Advances in Neural Information Processing Systems, 2018.
> >
> > [3] Sheng Shen, Zhewei Yao, Amir Gholami, Michael W. Mahoney, Kurt Keutzer, PowerNorm: Rethinking Batch Normalization in Transformers, Proceedings of the 37th International Conference on Machine Learning, PMLR, 2020

---

> > > ### Comment · Reviewer_mvwR · 2022-08-08
> > > **Thank you for the response!**
> > >
> > > Thank you for the response and the revised manuscript. The rebuttal partially addresses my concerns. I have increased my score to reflect that. My main remaining concern is regarding evaluation with multiple seeds. While it is nice to see that the results hold for an additional seed it would be more convincing to show all results averaged across multiple different seeds (e.g., 5).

---

> > > > ### Author Response · Authors · 2022-08-09
> > > > **Thank you for your reply!**
> > > >
> > > > We sincerely appreciate your positive evaluation of our response. We changed a sentence in the limitation section a bit so that we emphasize the importance of studying the average result of many more seeds.

---

### Author Response · Authors · 2022-08-01
**Common Response to All Reviewers**

Thank you very much all reviewers for taking time out of your valuable time to review our work. We have revised the manuscript to reflect your feedback and uploaded it. Changes in the revised manuscript are colored in red. We summarize below the main changes in the revised manuscript:

- We have moved the table showing baseline results from Appendix 1 to Section 4 of the main text (Table 2 → Table 1).
- We have moved the figure and description of CKA between different models from Section 5.1 to the Appendix (Fig. 2 → Fig. 12).
- We have added the takeaways of Section 5 at the end of the section.
- We have added a description of the dataset we used in Appendix B.
- We have added a new analysis of gradient norm in Appendix G.3.
- We have added an analysis of why the randomly initialized model fails for Hopper with no context in Appendix H.2.
- We have added the results of another seed (seed = 42) that we were able to conduct during this rebuttal period in Appendix J: CKA analysis, mutual information analysis, parameter similarity analysis, gradient analysis, block replacement analysis, and attention distance analysis.
- We have added notes on areas that may be difficult to understand in interpreting the results of the analysis in Appendix N, O, and P.

We have made other revisions to the wording and typo and added notes so that we reflect the results of our additional analysis and the points raised in the review. We will provide a more detailed explanation of the points raised by each reviewer separately to each of them. In the following, unless otherwise noted, figure and section indexes are those of the revised manuscript (e.g., Fig. 10 in the previous manuscript is now Fig. 9 in the new manuscript).

We hope that we have been able to improve the manuscript by incorporating your feedback. We would like to offer my sincerest thanks again to you all reviewers.

---

### Meta-Review · Area_Chair_55ic · 2022-08-24

**Recommendation:** Accept
**Confidence:** Certain

**Metareview:**

The paper unanimously receives positive rates thanks to strong motivations and interesting results. As the reviews show satisfaction on the authors’ feedback, the final draft needs to respect it accordingly, for example, about the limitations of this research.

**Award:**

No

---

### Decision · Program_Chairs · 2022-09-14

Accept